# Cross-Fitted Clipped Covariance Estimation with a Data-Driven Tail-Energy Criterion

## Abstract

Heavy-tailed data make covariance estimation sensitive to the clipping level: stronger clipping reduces variance but increases bias. We study how to choose this clipping level from the data within a radial clipped covariance family. We propose the quantile tail-energy surrogate (QTES), a fully data-driven rule that combines a cross-fitted variance certificate with a held-out estimate of the tail energy removed by clipping. QTES requires no distributional prior parameters. For Euclidean clipping, the operator-norm bias is bounded by this scalar tail-energy quantity. Under a finite $L_4$ moment condition, together with a mild feasibility condition for the QTES block construction, we prove a uniform finite-sample calibration bound over a positive clipping grid and a finite-sample guarantee for the estimator selected by QTES, relative to the best candidate on that grid under the same variance-plus-bias criterion. Experiments on clean heavy-tailed and contaminated benchmarks show that QTES performs strongly across both regimes and remains competitive with the best reference methods considered.

## 1 Introduction

This paper studies a specific tuning problem. We start with a simple robust covariance estimator that radially clips each observation, and we ask how to choose the clipping level from the data. We do not claim a new rate-optimal direct covariance estimator. The contribution is a data-driven rule for selecting the clipping level inside this clipped family.

Covariance estimation in operator norm is a basic ingredient in principal component analysis, factor modeling, and high-dimensional spectral inference (Johnstone, 2001; Paul, 2007; Fan et al., 2013). The standard starting point is the sample covariance matrix (SCM), which averages the rank-one matrices $Z_i Z_i^\top$. Many familiar procedures regularize this empirical second moment, but the raw average remains the object whose spectral behavior has to be controlled.

This average is fragile when the data are heavy-tailed or when a small fraction of observations are contaminated. A single observation with very large norm contributes the rank-one matrix $Z_i Z_i^\top$, whose size scales as $\|Z_i\|_2^2$; a few such observations can therefore dominate the largest eigenvalues and distort the leading eigenspaces. In this failure mode the covariance target may be well defined, but the empirical covariance is driven by rare extremes rather than by the typical geometry of the distribution. Under weak moment assumptions, this sensitivity to rare large observations becomes a dominant practical issue (Srivastava & Vershynin, 2013).

A direct way to reduce this instability is clipping. Before forming $Z_i Z_i^\top$, one replaces $Z_i$ by a radially clipped vector

$$\tilde{Z}_i(\tau) := \min\{1, \tau/\|Z_i\|_2\} Z_i,$$

so that no single rank-one contribution can exceed the scale $\tau^2$. Clipping makes the estimator less variable and less sensitive to outliers, but it introduces bias: if $\tau$ is too small, genuine covariance mass in the tails is removed; if $\tau$ is too large, the estimator behaves too much like the unstable sample covariance. The practical question is therefore how much to clip.

This paper focuses on that calibration problem. The candidate estimators are easy to compute once $\tau$ is fixed. The hard part is choosing $\tau$ without knowing the tail scale of the distribution.

The main observation is simple. For Euclidean clipping, the matrix bias created by clipping can be bounded by a one-dimensional tail quantity:

$$\Sigma - \mathbb{E}_P[\tilde{Z}(\tau)\tilde{Z}(\tau)^\top] = \mathbb{E}_P\Big[\Big(1 - \min\{1, \tau^2/\|Z\|_2^2\}\Big)ZZ^\top\Big] \succeq 0.$$

Therefore,

$$\big\|\Sigma - \mathbb{E}_P[\tilde{Z}(\tau)\tilde{Z}(\tau)^\top]\big\|_{\mathrm{op}} \leq \mathrm{tr}\big(\Sigma - \mathbb{E}_P[\tilde{Z}(\tau)\tilde{Z}(\tau)^\top]\big) = \mathbb{E}_P\big[(\|Z\|_2^2 - \tau^2)_+\big].$$

Thus, the matrix bias is controlled by a scalar tail-energy functional. This suggests a simple tuning principle. For each candidate clipping level, estimate two quantities from the data: how much random fluctuation remains after clipping, and how much tail energy has been removed by clipping. Then choose the candidate with the smallest estimated sum of these two quantities. We call this rule the quantile tail-energy surrogate (QTES). Its goal is fully data-driven calibration: choose a useful clipping level without any prior distributional parameter from the data-generating law.

**Related work and calibration gap.** For centered Gaussian sampling, the SCM has the usual effective-rank operator-norm benchmark $\|\Sigma\|_{\mathrm{op}}\sqrt{r(\Sigma)/n}$, suppressing constants, logarithmic factors, and confidence terms, where $r(\Sigma) := \mathrm{tr}(\Sigma)/\|\Sigma\|_{\mathrm{op}}$ (Lounici, 2014; Dirksen, 2015; Koltchinskii & Lounici, 2017). Robust covariance estimation asks for analogous control under weaker moments or contamination. Classical robust scatter estimators such as Maronna's estimator, Tyler's estimator, MCD, and MVE remain basic references under elliptical or shape-based modeling assumptions (Maronna, 1976; Tyler, 1987; Hubert et al., 2008). Modern nonasymptotic approaches include PAC-Bayesian and robust matrix-mean estimators (Catoni, 2016; Catoni & Giulini, 2017; Minsker, 2018), entrywise and contamination-focused estimators (Avella-Medina et al., 2018; Chen et al., 2018; Minsker & Wang, 2024), and direct robust covariance estimators with effective-rank guarantees under moment, norm-equivalence, or contamination assumptions (Wei & Minsker, 2017; Mendelson & Zhivotovskiy, 2020; Oliveira & Rico, 2024; Abdalla & Zhivotovskiy, 2026).

This leaves a practical calibration gap. Some optimal-rate robust covariance constructions are algorithmic only at a high level, computationally expensive, or not directly implementable as a simple finite-sample routine. The estimators that are simple and computable, especially truncation and clipping procedures, typically require tuning parameters involving distributional quantities such as a tail scale, moment-comparison constant, covariance norm, effective-rank scale, or contamination level. In simulations one can supply these quantities as oracle inputs, but in real data analysis they are parameters of the unknown distribution that the covariance estimator is meant to learn.

This motivates the user-friendly calibration viewpoint of Ke et al. (2019): robust covariance methods should be computable and tunable from the observed sample, rather than from unavailable distributional constants. Following this viewpoint for Euclidean radial clipping, QTES selects the clipping level without prior distributional parameters and addresses the data-driven clipping-level selection problem inside a simple computable family. The resulting guarantee is a selector guarantee for the clipped family, not a sharp direct effective-rank rate; the comparison with such rates is given in Section 2.4.

The closest technical line is truncation and clipping. The rank-one truncation estimator of Wei & Minsker (2017) is algebraically the same as Euclidean radial clipping in the centered single-scale case; Section 3.1 spells out this equivalence when defining the WM-oracle benchmark. The U-statistic refinements of Minsker & Wei (2020) and the contamination and low-rank extensions of Minsker & Wang (2024) develop related robust covariance mechanisms. The robust mean estimators that motivate our bias upper-confidence-bound construction trace back to Huber (1964); Catoni (2012); Devroye et al. (2016); Lugosi & Mendelson (2019), and the Huber-style adaptive calibration viewpoint is also in the spirit of Sun et al. (2020); Wang et al. (2021).

**Contributions.**

- We isolate the tail-energy identity for Euclidean clipping and reduce operator-norm bias control to a scalar tail-energy problem.

- We propose QTES, a fully data-driven calibration rule that combines a cross-fitted variance certificate with a held-out bias surrogate. It selects the clipping level by grid search and requires no prior distributional parameters such as a tail scale, moment constant, covariance norm. Under the $L_4$ condition, together with the block-feasibility requirement for the QTES construction, the selector satisfies a uniform finite-sample calibration bound and a near-oracle inequality on the positive clipping grid.

- Empirically, QTES improves over uninformed clipping choices and is compared with SCM, oracle approximating shrinkage (OAS), minimum covariance determinant (MCD), `tfHuber` (Dai & Sun, 2021; Ke et al., 2019), Minsker–Wei robust $U$-statistic diagnostics (Minsker & Wei, 2020), and a scale-oracle Wei–Minsker benchmark (Wei & Minsker, 2017), which anchors the comparison to an optimal-rate truncation method when its unknown scale is supplied.

**Data model.** Let $P$ denote the underlying distribution. We observe independent samples $Z_1, \ldots, Z_n \sim P$ in $\mathbb{R}^d$ with $\mathbb{E}_P[Z_1] = 0$ and target covariance $\Sigma = \mathbb{E}_P[Z_1 Z_1^\top]$. We state the main result in the centered setting to focus on clipping-level selection. Appendix D explains how to add foldwise mean estimation when the mean is unknown. We assume the finite fourth-moment condition

$$\mathbb{E}_P \|Z_1\|_2^4 < \infty. \tag{1}$$

**Notation.** For a vector $v \in \mathbb{R}^d$, $\|v\|_2$ denotes its Euclidean norm. For a symmetric matrix $A \in \mathbb{R}^{d \times d}$, $\|A\|_{\mathrm{op}}$ denotes its operator norm, $\mathrm{tr}(A)$ its trace, and $A \succeq 0$ indicates that $A$ is positive semidefinite (PSD). For a scalar $x \in \mathbb{R}$, $(x)_+ := \max\{x, 0\}$ denotes the positive part. For a finite set $S$, $|S|$ denotes its cardinality. We write $\mathbb{E}_P$ and $\mathrm{Var}_P$ for expectation and variance under the underlying distribution $P$. When the distribution is clear, we also write $\mathbb{E}$. The indicator function of an event $\mathcal{E}$ is denoted by $\mathbf{1}\{\mathcal{E}\}$. For a collection of random variables $X$, $\sigma(X)$ denotes the $\sigma$-algebra generated by $X$. We write $a \lesssim b$ when $a \leq Cb$ for a universal constant $C$.

## 2 Method and Main Results

This section constructs the estimator and states the main guarantee. The organizing idea is the variance–bias tradeoff created by clipping. Strong clipping lowers variance because large rank-one contributions are capped, but it raises bias because true tail covariance is removed. Weak clipping has the opposite effect. QTES chooses the clipping level by estimating both sides of this tradeoff from the data.

The construction follows this order. Section 2.1 handles the variance side: it defines the cross-fitted clipped covariance family and builds a computable variance certificate for each grid point. Section 2.2 handles the bias side: it shows that the clipping bias is bounded by a scalar tail-energy quantity and estimates that quantity from held-out data. Section 2.3 puts the two sides together: it defines the QTES variance-plus-bias score, selects the grid point with the smallest score, and states the finite-sample guarantee for the selected estimator. Section 2.4 explains how to interpret the selector guarantee relative to optimal covariance rates: QTES is a data-driven selector for a clipped estimator family, and the theorem controls the calibration remainder, while the in-grid oracle term reflects the approximation quality of that family.

### 2.1 Cross-fitted clipping

This subsection handles the variance side of the tradeoff. Its output is a collection of clipped covariance estimates, one for each grid point $\gamma \in \mathcal{G}$, together with a computable variance bound for each one. The main device is cross-fitting: for each fold, the other fold chooses the clipping radius, and the held-out fold is used to form the covariance estimate and certify its fluctuation. This separation makes the radius fixed after conditioning on the training fold, which is what allows the matrix concentration bound to be applied.

The construction has three steps. First, we split the sample into two equal even folds; even fold sizes are needed because the variance proxy pairs observations. Second, we define the grid of clipping fractions. Third, for every grid point and fold, we compute a training-fold radius and a held-out clipped covariance estimate.

Let $\Pi$ denote a random permutation of the sample indices drawn independently of the data values. We use $\Pi$ to retain the first $n_{\text{ret}} := 4\lfloor n/4 \rfloor$ indices, and split them into two disjoint test folds $I_1$ and $I_2$ of equal size. The factor four is used so that each test fold has even size; the variance proxy below pairs observations inside each fold. The corresponding training folds are $J_1 := I_2$ and $J_2 := I_1$. Write $n_k := |I_k|$, so $n_1 = n_2 = n_{\text{ret}}/2$.

The clipping radius on fold $k$ is random because it is computed from the training fold $J_k$. To apply concentration on the held-out fold, we condition on all information used to choose this radius:

$$\mathcal{F}_k := \sigma(\Pi, \{Z_i\}_{i \in J_k}).$$

Conditional on $\mathcal{F}_k$, the grid and the radii $\tau_k(\gamma)$ are fixed, while the observations in $I_k$ remain independent. This is the role of the filtration.

We tune over a finite geometric grid of clipping fractions. Here $\gamma$ is a clipping fraction rather than a radius: roughly a $\gamma$ fraction of the training-fold radii lie above the chosen threshold. Thus larger $\gamma$ means a smaller clipping radius and more aggressive clipping.

Fix a ratio $\rho > 1$ and set

$$\gamma_{\max} := 0.3, \qquad \gamma_{\min} := \min\left\{0.25, \frac{1}{\min_k |J_k|}\right\}.$$

Let

$$J_{\max} := \left\lfloor \frac{\log(\gamma_{\max}/\gamma_{\min})}{\log \rho} \right\rfloor.$$

The grid is

$$\mathcal{G} := \{\gamma_j = \gamma_{\max} \rho^{-j} : j = 0, \ldots, J_{\max}\}. \tag{2}$$

Thus $\mathcal{G}$ is fixed once $\rho$ and the fold sizes are fixed. In the experiments we use the dyadic spacing $\rho = 2$.

For $\gamma \in \mathcal{G}$, fold $k$, and the ordered training radii $R_{(1)}^{(k)} \leq \cdots \leq R_{(|J_k|)}^{(k)}$, define $\tau_k(\gamma) := R_{(|J_k|-q_{k,\gamma})}^{(k)}$ where $q_{k,\gamma} := \lfloor \gamma |J_k| \rfloor$. In words, $q_{k,\gamma}$ is the number of largest training-fold radii that are allowed to lie above the threshold. A held-out observation is shrunk only when its norm exceeds this threshold. On the test fold $I_k$, the clipped covariance is

$$\hat{\Sigma}_k^{\text{raw}}(\gamma) := \frac{1}{n_k} \sum_{i \in I_k} \tilde{Z}_i^{(k)}(\gamma) \tilde{Z}_i^{(k)}(\gamma)^\top, \qquad \tilde{Z}_i^{(k)}(\gamma) := \begin{cases} Z_i \min\{1, \tau_k(\gamma)/\|Z_i\|_2\}, & \|Z_i\|_2 > 0, \\ 0, & \|Z_i\|_2 = 0. \end{cases} \tag{3}$$

with aggregate estimator $\hat{\Sigma}^{\text{raw}}(\gamma) = \sum_{k=1}^{2} \frac{n_k}{n_{\text{ret}}} \hat{\Sigma}_k^{\text{raw}}(\gamma)$.

We next build the variance part of the score. After conditioning on $\mathcal{F}_k$, the clipped matrices on $I_k$ are independent and bounded, so a matrix empirical-Bernstein bound can be applied. Define the normalized matrices $A_i^{(k)}(\gamma) := \tilde{Z}_i^{(k)}(\gamma) \tilde{Z}_i^{(k)}(\gamma)^\top / \tau_k(\gamma)^2$ for $\tau_k(\gamma) > 0$, and set $A_i^{(k)}(\gamma) = 0$ when $\tau_k(\gamma) = 0$. The normalization by $\tau_k(\gamma)^2$ puts each clipped rank-one matrix on the common scale $0 \preceq A_i^{(k)}(\gamma) \preceq I_d$, which is the bounded matrix input required by the empirical-Bernstein inequality. Following Wang & Ramdas (2025), fix any ordering $I_k = \{i_{k,1}, \ldots, i_{k,n_k}\}$. The paired differences below estimate the conditional matrix variance without requiring the unknown conditional mean of the clipped matrices:

$$V_k^\star(\gamma) := \frac{1}{n_k} \sum_{j=1}^{n_k/2} \left(A_{i_{k,2j-1}}^{(k)}(\gamma) - A_{i_{k,2j}}^{(k)}(\gamma)\right)^2, \tag{4}$$

The corresponding matrix empirical-Bernstein radius is

$$D_{m,d}(\alpha; V) := \frac{\log\left(\frac{md}{(m-1)\alpha}\right)}{3m} + \sqrt{\frac{2\|V\|_{\text{op}} \log\left(\frac{md}{(m-1)\alpha}\right)}{m}} + \left(\sqrt{\frac{5}{3}} + 1\right) \frac{\sqrt{\log\left(\frac{md}{(m-1)\alpha}\right) \log\left(\frac{2md}{\alpha}\right)}}{m}. \tag{5}$$

The expression $D_{m,d}$ should be read as a computable empirical-Bernstein radius: its leading term is of order $\sqrt{\|V\|_{\text{op}} \log(d/\alpha)/m}$, with lower-order logarithmic corrections.

Using $\alpha_\gamma := \delta_{\mathrm{var}}/(4|\mathcal{G}|)$, define

$$\Psi_k^{\mathrm{raw}}(\gamma) := \tau_k(\gamma)^2 D_{n_k,d}(\alpha_\gamma; V_k^\star(\gamma)), \qquad \Psi^{\mathrm{raw}}(\gamma) := \sum_{k=1}^{2} \frac{n_k}{n_{\mathrm{ret}}} \Psi_k^{\mathrm{raw}}(\gamma). \tag{6}$$

Over the increasing grid $\gamma^{(1)} < \cdots < \gamma^{(|\mathcal{G}|)}$, define the suffix maximum $\overline{\Psi}^{\mathrm{raw}}(\gamma^{(j)}) := \max_{t \geq j} \Psi^{\mathrm{raw}}(\gamma^{(t)})$. This is a monotonicity adjustment: since more aggressive clipping should not increase the variance scale, the suffix maximum enforces this direction on the computed certificate.

**Theorem 1** (Uniform variance certificate). *Assume $Z_1, \ldots, Z_n$ are independent and identically distributed (i.i.d.) from $P$. Define*

$$\bar{\Sigma}_P^{\mathrm{raw}}(\gamma) := \sum_{k=1}^{2} \frac{n_k}{n_{\mathrm{ret}}} \mathbb{E}_P\big[\hat{\Sigma}_k^{\mathrm{raw}}(\gamma) \mid \mathcal{F}_k\big].$$

*This is the fold-conditional mean of the clipped estimator. It is an analysis object; the algorithm never computes it. With probability at least $1 - \delta_{\mathrm{var}}$, simultaneously for all $\gamma \in \mathcal{G}$,*

$$\left\| \hat{\Sigma}^{\mathrm{raw}}(\gamma) - \bar{\Sigma}_P^{\mathrm{raw}}(\gamma) \right\|_{\mathrm{op}} \leq \overline{\Psi}^{\mathrm{raw}}(\gamma). \tag{7}$$

Theorem 1 says that the random fluctuation of each clipped estimator around its fold-conditional mean is bounded by a quantity computed from the held-out clipped data. The bound holds for all grid points at once.

The remaining error is the bias caused by clipping. For each fold $k$ and grid point $\gamma$, let $Z_k^{\mathrm{test}} \sim P$ be independent of $\mathcal{F}_k$, and define

$$Y_k^{\mathrm{test}}(\gamma) := (\|Z_k^{\mathrm{test}}\|_2^2 - \tau_k(\gamma)^2)_+.$$

The symbol $Z_k^{\mathrm{test}}$ is a notational device for a fresh held-out draw under the conditional radius. Conditional on $\mathcal{F}_k$, $Y_k^{\mathrm{test}}(\gamma)$ has the same law as any $Y_i^{(k)}(\gamma)$ with $i \in I_k$.

## 2.2 Tail-energy score

This subsection handles the bias side of the tradeoff. The variance certificate from Section 2.1 controls random fluctuation around the fold-conditional mean, but that conditional mean is still biased because clipping removes radial mass. We first show that this matrix bias is bounded by a scalar tail-energy quantity. We then estimate that scalar quantity from the held-out fold using block means and an upper quantile.

The argument uses a property specific to Euclidean clipping: the removed covariance mass is positive semidefinite, so its operator norm is no larger than its trace. That trace is the expected squared norm removed beyond the clipping radius.

**Lemma 2** (Tail-energy representation of the bias). *Fix a fold $k$ and a grid point $\gamma$, and let*

$$B_k(\gamma) := \Sigma - \mathbb{E}_P\big[\hat{\Sigma}_k^{\mathrm{raw}}(\gamma) \mid \mathcal{F}_k\big].$$

*Thus $B_k(\gamma)$ is the population covariance mass lost by clipping on fold $k$. Then $B_k(\gamma) \succeq 0$ almost surely and*

$$\|B_k(\gamma)\|_{\mathrm{op}} \leq \mathrm{tr}(B_k(\gamma)) = \mathbb{E}_P\big[Y_k^{\mathrm{test}}(\gamma) \mid \mathcal{F}_k\big]. \tag{8}$$

*Consequently, for*

$$b(\gamma) := \sum_{k=1}^{2} \frac{n_k}{n_{\mathrm{ret}}} \mathbb{E}_P\big[Y_k^{\mathrm{test}}(\gamma) \mid \mathcal{F}_k\big],$$

*we have $\left\| \Sigma - \bar{\Sigma}_P^{\mathrm{raw}}(\gamma) \right\|_{\mathrm{op}} \leq b(\gamma)$.*

We estimate this scalar bias term from the held-out fold. For each fold $k$ and grid point $\gamma$, define the held-out tail-excess variables

$$Y_i^{(k)}(\gamma) := (\|Z_i\|_2^2 - \tau_k(\gamma)^2)_+, \qquad i \in I_k. \tag{9}$$

Conditional on $\mathcal{F}_k$, the variables $\{Y_i^{(k)}(\gamma)\}_{i \in I_k}$ are i.i.d., each with the same conditional law as $Y_k^{\text{test}}(\gamma)$. We use block means and an upper quantile of those block means to estimate the tail-energy bias from above. The block means reduce the influence of a few extreme tail-excess values.

We split the confidence budget as $\delta_{\text{var}} = \delta_{\text{bias}} = \delta/2$ and define the baseline number of blocks by

$$m_{\text{base}} := \left\lceil 4 \log\left(\frac{4|\mathcal{G}|}{\delta_{\text{bias}}}\right) \right\rceil. \tag{10}$$

Set $m_{\text{blocks}} := \min\{m_{\text{base}}, n_1\}$. For the theorem below we impose the block-feasibility condition

$$m_{\text{base}} \le n_1, \tag{11}$$

which, because $n_1 = n_2$ by construction, guarantees that every block on each fold contains at least one sample and makes $m_{\text{blocks}} = m_{\text{base}}$. This is a sample-size requirement for the block construction, not an additional distributional assumption. For each $(k, \gamma)$, we independently shuffle $\{Y_i^{(k)}(\gamma)\}_{i \in I_k}$, partition them into $m_{\text{blocks}}$ groups whose sizes differ by at most one, and let $\widehat{b}_k^{\text{QTES}}(\gamma)$ be the empirical 85% quantile of the resulting block means. We then set

$$\widehat{b}^{\text{QTES}}(\gamma) := \sum_{k=1}^2 \frac{n_k}{n_{\text{ret}}} \widehat{b}_k^{\text{QTES}}(\gamma), \qquad \widehat{S}_{\text{QTES}}(\gamma) := \overline{\Psi}^{\text{raw}}(\gamma) + \widehat{b}^{\text{QTES}}(\gamma), \qquad \hat{\gamma}_{\text{QTES}} \in \arg\min_{\gamma \in \mathcal{G}} \widehat{S}_{\text{QTES}}(\gamma). \tag{12}$$

The score $\widehat{S}_{\text{QTES}}(\gamma)$ is the estimated variance plus the estimated clipping bias. Large $\gamma$ clips more aggressively, lowering the variance term at the cost of increasing the bias term; small $\gamma$ does the reverse. The theorem below states the resulting guarantee for the QTES-selected estimator.

## 2.3 Main result

This subsection combines the variance and bias sides into a selector. The QTES score is designed to approximate the ideal variance-plus-bias curve

$$\overline{\Psi}^{\text{raw}}(\gamma) + b(\gamma),$$

where $\overline{\Psi}^{\text{raw}}(\gamma)$ is the variance certificate and $b(\gamma)$ is the conditional clipping bias bound. The theorem says that, under a finite fourth moment and the block-feasibility condition, the data-driven score is uniformly close to this ideal curve over the grid. As a result, minimizing the QTES score returns an estimator whose error is close to the best grid candidate under that same ideal criterion. This is a selector guarantee inside the clipped family.

Define

$$s_\star := \left\lfloor \frac{n_1}{m_{\text{blocks}}} \right\rfloor, \qquad M_4 := \left(\mathbb{E}_P \|Z_1\|_2^4\right)^{1/4}, \qquad \Delta_n := 10 M_4^2 \, s_\star^{-1/2}.$$

Here $s_\star$ is the smallest possible block size, $M_4$ is the global fourth-moment scale, and $\Delta_n$ is the price paid for calibrating the tail-energy score uniformly over the grid.

**Theorem 3** (Uniform QTES bound and selected-estimator guarantee). *Assume* (1) *and* (11). *Then with probability at least* $1 - \delta$, *simultaneously for all* $\gamma \in \mathcal{G}$,

$$\left|\widehat{b}^{\text{QTES}}(\gamma) - b(\gamma)\right| \le \Delta_n, \qquad \left\|\widehat{\Sigma}^{\text{raw}}(\gamma) - \Sigma\right\|_{\text{op}} \le \widehat{S}_{\text{QTES}}(\gamma) + \Delta_n. \tag{13}$$

*Consequently,*

$$\left\|\widehat{\Sigma}^{\text{raw}}(\hat{\gamma}_{\text{QTES}}) - \Sigma\right\|_{\text{op}} \le \min_{\gamma \in \mathcal{G}} \left\{\overline{\Psi}^{\text{raw}}(\gamma) + b(\gamma)\right\} + 2\Delta_n. \tag{14}$$

The first inequality says that the observed QTES bias estimate tracks the ideal conditional bias term. The second says that the QTES score is a simultaneous upper bound on the actual covariance error, up to the same calibration slack. Appendix B gives the proof in two steps: a generic bias-upper-bound oracle inequality,

followed by the calibration argument showing that the QTES block-quantile estimate satisfies the required bias bound.

The next corollary states the corresponding fact about the selected grid point: QTES nearly minimizes the ideal curve $O(\gamma) = \overline{\Psi}^{\mathrm{raw}}(\gamma) + b(\gamma)$, and it selects the exact minimizer if the gap between the best grid point and the rest is larger than the calibration error.

**Corollary 4** (Near-optimal selected candidate for the ideal curve). *Define*

$$O(\gamma) := \overline{\Psi}^{\mathrm{raw}}(\gamma) + b(\gamma), \qquad \gamma \in \mathcal{G}.$$

*Under the same event as Theorem 3,*

$$O(\hat{\gamma}_{\mathrm{QTES}}) \leq \min_{\gamma \in \mathcal{G}} O(\gamma) + 2\Delta_n. \tag{15}$$

*Hence for every $\varepsilon \geq 2\Delta_n$, the selected grid point belongs to the $\varepsilon$-near-minimizer set*

$$\Gamma_\varepsilon := \{\gamma \in \mathcal{G} : O(\gamma) \leq \min_{t \in \mathcal{G}} O(t) + \varepsilon\}.$$

*In particular, if $\gamma_\star$ is a unique minimizer of $O$ and*

$$\min_{\gamma \in \mathcal{G} \setminus \{\gamma_\star\}} \big(O(\gamma) - O(\gamma_\star)\big) > 2\Delta_n,$$

*then $\hat{\gamma}_{\mathrm{QTES}} = \gamma_\star$.*

The proofs of Theorem 3 and Corollary 4 are deferred to Appendix C.

Algorithm 1 gives the full procedure. In words, QTES first splits the retained sample into two folds. For each candidate clipping fraction $\gamma$, each fold uses the other fold to choose a clipping radius. It then clips the held-out fold and computes two quantities there: a variance certificate for the clipped covariance and a block-quantile estimate of the tail energy removed by clipping. These two quantities are added to form the QTES score for $\gamma$. The algorithm repeats this over the finite grid, selects the $\gamma$ with the smallest score, and returns the corresponding cross-fitted clipped covariance estimate.

---

**Algorithm 1** Cross-fitted QTES on the positive clipping grid

---

**Require:** Centered samples $Z_1, \ldots, Z_n \in \mathbb{R}^d$, confidence $\delta \in (0,1)$, grid ratio $\rho = 2.0$.
**Ensure:** Tuned covariance estimate $\hat{\Sigma}^{\mathrm{raw}}(\hat{\gamma}_{\mathrm{QTES}})$.
1: Set $\delta_{\mathrm{var}} = \delta_{\mathrm{bias}} = \delta/2$.
2: Retain $n_{\mathrm{ret}} = 4\lfloor n/4 \rfloor$ observations; split evenly into test folds $I_1, I_2$; define training folds $J_1 = I_2$, $J_2 = I_1$.
3: Set $\gamma_{\max} = 0.3$, $\gamma_{\min} = \min\{0.25, 1/\min_k |J_k|\}$, and build $\mathcal{G}$ by (2).
4: Compute $m_{\mathrm{base}} = \lceil 4\log(4|\mathcal{G}|/\delta_{\mathrm{bias}}) \rceil$ and set $m_{\mathrm{blocks}} = \min\{m_{\mathrm{base}}, n_1\}$.
5: **for** $\gamma \in \mathcal{G}$ **do**
6:     **for** $k \in \{1, 2\}$ **do**
7:         Compute $\tau_k(\gamma)$ from the training-fold radii and assemble $\hat{\Sigma}_k^{\mathrm{raw}}(\gamma)$ via (3).
8:         Compute $V_k^\star(\gamma)$ and $\Psi_k^{\mathrm{raw}}(\gamma)$ via (4)–(6).
9:         Form $Y_i^{(k)}(\gamma) = (\|Z_i\|_2^2 - \tau_k(\gamma)^2)_+$ for $i \in I_k$.
10:        Randomly partition $I_k$ into $m_{\mathrm{blocks}}$ blocks and compute the mean of $Y$ on each block.
11:        Set $\hat{b}_k^{\mathrm{QTES}}(\gamma)$ to the empirical 85% quantile of these block means.
12:     **end for**
13:     Aggregate $\Psi^{\mathrm{raw}}(\gamma)$ and $\hat{b}^{\mathrm{QTES}}(\gamma)$.
14: **end for**
15: Monotonize the variance radius: $\overline{\Psi}^{\mathrm{raw}}(\gamma^{(j)}) = \max_{t \geq j} \Psi^{\mathrm{raw}}(\gamma^{(t)})$.
16: Select $\hat{\gamma}_{\mathrm{QTES}} \in \arg\min_{\gamma \in \mathcal{G}} \hat{S}_{\mathrm{QTES}}(\gamma)$.
17: Return $\hat{\Sigma}^{\mathrm{raw}}(\hat{\gamma}_{\mathrm{QTES}})$.

---

## 2.4 Relation to optimal rates

QTES is analyzed as a selector for a clipped family, not as a new direct estimator with an optimal rate. Let $r(\Sigma) := \mathrm{tr}(\Sigma)/\|\Sigma\|_{\mathrm{op}}$. Suppressing constants, logarithmic factors, and confidence terms, the optimal rate or

effective-rank rate for operator-norm covariance estimation is

$$\|\Sigma\|_{\mathrm{op}}\sqrt{\frac{r(\Sigma)}{n}}.$$

Several robust direct estimators attain this order, up to the suppressed factors, under suitable moment or norm-equivalence assumptions (Wei & Minsker, 2017; Mendelson & Zhivotovskiy, 2020; Minsker & Wei, 2020; Oliveira & Rico, 2024; Abdalla & Zhivotovskiy, 2026).

By contrast, our theorem yields

$$\left\|\hat{\Sigma}^{\mathrm{raw}}(\hat{\gamma}_{\mathrm{QTES}}) - \Sigma\right\|_{\mathrm{op}} \le \min_{\gamma\in\mathcal{G}} O(\gamma) + 2\Delta_n, \qquad \Delta_n = 10\big(\mathbb{E}_P\|Z_1\|_2^4\big)^{1/2} s_\star^{-1/2}.$$

The oracle term $\min_{\gamma\in\mathcal{G}} O(\gamma)$ depends on how well the clipped family can approximate $\Sigma$ at the available grid points. The part controlled by the present theorem, and therefore the part to compare with the direct rates above, is the calibration remainder $\Delta_n$. Since $s_\star$ is of order $n/\log(|\mathcal{G}|/\delta)$, and since the same $L_4$–$L_2$ comparison gives

$$\big(\mathbb{E}\|Z\|_2^4\big)^{1/2} \lesssim \mathrm{tr}(\Sigma) = r(\Sigma)\|\Sigma\|_{\mathrm{op}},$$

the QTES selector remainder has the rough order

$$\Delta_n \lesssim r(\Sigma)\|\Sigma\|_{\mathrm{op}}\sqrt{\log(|\mathcal{G}|/\delta)/n} = \sqrt{r(\Sigma)}\,\|\Sigma\|_{\mathrm{op}}\sqrt{r(\Sigma)\log(|\mathcal{G}|/\delta)/n},$$

This remainder is larger than the sharp direct effective-rank scale by a factor of order $\sqrt{r(\Sigma)}$, apart from logarithmic terms. The comparison is only for the calibration remainder: the theorem still contains the oracle term $\min_{\gamma\in\mathcal{G}} O(\gamma)$. This looseness reflects the trace-tail relaxation used by the selector guarantee.

As the experiments below illustrate, the looseness of this calibration bound does not preclude strong practical performance.

## 3 Experiments

This section checks the selector in simulation. We ask three questions. First, does QTES choose a clipping level close to the best grid point in hindsight? Second, how does the resulting covariance estimator compare with conventional covariance estimators and robust reference methods on clean and contaminated data? Third, how does the oracle-approximation gap of the data-driven selector change as $n$ grows? Unless stated otherwise, all reported table entries are averages over repeated Monte Carlo trials. The primary metric is the relative spectral error

$$\mathrm{RelSpecErr}(\hat{\Sigma}, \Sigma) := \frac{\|\hat{\Sigma} - \Sigma\|_{\mathrm{op}}}{\|\Sigma\|_{\mathrm{op}}}.$$

For eigenspace recovery we also report

$$\mathrm{SubspaceErr}(\hat{\Sigma}, \Sigma; s) := \|P_s(\hat{\Sigma}) - P_s(\Sigma)\|_{\mathrm{op}},$$

where $P_s(A)$ is the orthogonal projector onto the leading $s$-dimensional eigenspace of $A$. By the Davis–Kahan theorem (Yu et al., 2015), this subspace distance is fundamentally controlled by the operator-norm error of the covariance estimate provided there is a sufficient eigengap. Runtime is mean wall-clock time per fit. We fix $\gamma_{\mathrm{max}} = 0.3$ and evaluate the positive grid $\mathcal{G}$ used in the theory.

### 3.1 Experimental setup

This subsection defines the common simulation model and the baselines. We keep the covariance matrix fixed across experiments so that changes in performance come from the distributional regime and the tuning problem.

The target covariance is a spiked model

$$\Sigma = Q\,\mathrm{diag}(1+\theta, \ldots, 1+\theta, 1, \ldots, 1)\,Q^\top$$

with rank $s = 5$, spike strength $\theta = 10$, and a single random orthogonal matrix $Q$ fixed across trials within each experiment. Data is generated as $Z_i = \Sigma^{1/2}\xi_i$, where $\Sigma^{1/2}$ denotes the symmetric positive semidefinite square root of $\Sigma$, and the coordinates of $\xi_i$ are standardized and independent. We consider three coordinate laws: Gaussian, Student-$t(5)$ rescaled to unit variance, and standardized lognormal. All synthetic distributions are mean zero, and throughout the main experiments in this section we treat this mean as known: the raw draws are passed to every estimator without empirical recentering. Appendix D reports a separate nonzero-mean check with foldwise mean estimation.

Unless noted otherwise, we set $(n, d) = (400, 200)$ for all experiments. The comparison includes QTES, SCM, OAS, MCD, two Wei–Minsker truncation references denoted WM-oracle and WM-adaptive (Wei & Minsker, 2017), two Minsker–Wei robust matrix $M$-estimator references denoted MW-Ustat-oracle and MW-Ustat-adaptive (Minsker & Wei, 2020), two `tfHuber` covariance modes, which belong to the tuning-free Huber covariance estimator denoted `tfHuber`-elem and `tfHuber`-spec (Dai & Sun, 2021; Ke et al., 2019). All benchmark rows are averages over 100 trials except `tfHuber`.

The oracle-assisted scale inputs are computed as follows.

For Wei & Minsker (2017), the scale is

$$\sigma_0 = \left\| \mathbb{E}\left[\|Z\|_2^2 ZZ^\top\right] \right\|_{\mathrm{op}}^{1/2}.$$

WM-oracle uses $\sigma_{\min} = \sigma_{\max} = \sigma_0$, so the grid has one point and the Lepski step is skipped. WM-adaptive is given the prior interval $[\sigma_0/2, 2\sigma_0]$.

For MW-Ustat, the scale is

$$\sigma_* = \left\| \mathbb{E}\left[(ZZ^\top - \Sigma)^2\right] \right\|_{\mathrm{op}}^{1/2}.$$

MW-Ustat-oracle sets $t = 1$ and uses the exact value $\sigma_*$, so it also skips Lepski selection. MW-Ustat-adaptive uses $\sigma_{\min} = \sigma_*/2$ and ratio $\gamma = 2$; we retain the first three scales $\{\sigma_*/2, \sigma_*, 2\sigma_*\}$. In the contamination experiments, all oracle scale inputs are computed from the clean lognormal inlier law, matching the clean target covariance. Following the numerical section of Minsker & Wei (2020), each MW-Ustat candidate is approximated by two gradient-descent steps.

The primary oracle benchmark is WM-oracle. In the centered rank-one case, the single-scale Wei–Minsker estimator is algebraically identical to Euclidean radial clipping:

$$\hat{\Sigma}_{\mathrm{WM}}(\theta) = \frac{1}{m}\sum_{i=1}^m Z_i Z_i^\top \min\left\{1, \frac{1}{\theta\|Z_i\|_2^2}\right\} = \frac{1}{m}\sum_{i=1}^m \tilde{Z}_i(\tau)\tilde{Z}_i(\tau)^\top, \qquad \tau^2 = 1/\theta.$$

Thus WM-oracle serves as an excellent performance benchmark for QTES: it is a scale-known member of the same clipped family, backed by the optimal-rate truncation theory of Wei & Minsker (2017).

## 3.2 Selector quality

This experiment focuses only on clipping-level selection. At $(n, d) = (400, 200)$, using 100 independent trials for each distribution, we compare the selected grid point with the best grid point in hindsight. The theory controls the ideal criterion

$$O(\gamma) = \overline{\Psi}^{\mathrm{raw}}(\gamma) + b(\gamma),$$

whereas Table 1 evaluates the clipping level selected from the finite grid $\mathcal{G}$ against the realized in-grid oracle under relative spectral error.

The columns are interpreted as follows. *QTES* is the mean relative spectral error achieved by the clipping level selected by our procedure. *Random-$\gamma$* is the expected error of a uniformly random choice from $\mathcal{G}$, implemented trialwise as the grid-average error. *Oracle* is the mean error of the best grid point in hindsight on each trial. *Sel./Oracle* is the ratio between the selected error and the oracle error, so values closer to 1 are better. *Top-2 Hit* is the fraction of trials in which the selected $\gamma$ lies among the two best grid points, so larger values are better.

Table 1: Selector diagnostics at $(n, d) = (400, 200)$ over 100 trials per distribution

| Distribution | QTES | Random-$\gamma$ | Oracle | Sel./Oracle | Top-2 Hit |
|---|---|---|---|---|---|
| Gaussian | 0.3330 | 0.3384 | 0.3122 | 1.0660 | 0.5000 |
| Student-$t(5)$ | 0.3335 | 0.3410 | 0.3130 | 1.0657 | 0.5000 |
| Lognormal | 0.3492 | 0.3713 | 0.3296 | 1.0585 | 0.5800 |

Overall, QTES consistently improves on the random baseline and stays close to the in-grid oracle across all three distributions: the selected/oracle ratio ranges from 1.0585 to 1.0660, and the Top-2 hit rate ranges from 0.50 to 0.58. The selector should therefore be interpreted as a stable oracle-approximation rule rather than an exact oracle-recovery mechanism.

### 3.3 Benchmark

We now evaluate the covariance estimator returned by QTES at $(n, d) = (400, 200)$. The goal is to compare QTES with reference methods under heavy-tailed sampling.

Table 2 compares the estimators introduced in the previous subsection under clean sampling. Within each distribution block, *RelSpecErr* is the relative operator-norm error, *SubspaceErr* is the operator-norm difference between the leading-$s$ eigenspace projectors, and *Time (s)* is the wall-clock time per fit. The first two columns measure statistical accuracy, while the last column records computational cost.

Table 2: Clean-data benchmark at $(n, d) = (400, 200)$

| Method | Gaussian | | | Student-$t(5)$ | | | Lognormal | | |
|---|---|---|---|---|---|---|---|---|---|
| | RelSpecErr | SubspaceErr | Time (s) | RelSpecErr | SubspaceErr | Time (s) | RelSpecErr | SubspaceErr | Time (s) |
| QTES | **0.3330** | 0.2625 | 0.0300 | **0.3335** | **0.2618** | 0.0303 | 0.3492 | **0.2768** | 0.0269 |
| SCM | 0.3534 | **0.2621** | 0.0004 | 0.3618 | 0.2640 | 0.0004 | 0.5994 | 0.3665 | 0.0004 |
| OAS | 0.3919 | **0.2621** | 0.0081 | 0.3909 | 0.2640 | 0.0081 | 0.5045 | 0.3665 | 0.0075 |
| MCD | 0.4101 | 0.3021 | 0.6514 | 0.4138 | 0.3045 | 0.6474 | 0.4139 | 0.3421 | 0.7239 |
| WM-oracle | 0.3487 | 0.2622 | 0.0007 | 0.3451 | 0.2621 | 0.0007 | **0.3417** | 0.2797 | 0.0007 |
| WM-adaptive | 0.5029 | 0.2654 | 0.0048 | 0.4952 | 0.2633 | 0.0114 | 0.4462 | 0.2648 | 0.0124 |
| MW-Ustat-oracle | 0.3686 | 0.2623 | 2.7985 | 0.3694 | 0.2618 | 2.6923 | 0.3823 | 0.2791 | 2.7190 |
| MW-Ustat-adaptive | 0.5506 | 0.2642 | 8.6761 | 0.5493 | 0.2622 | 8.3772 | 0.5274 | 0.2656 | 8.4246 |
| `tfHuber`-elem.[†] | 0.3133 | 0.2557 | 319.3 | 0.3593 | 0.2611 | 345.5 | 0.3248 | 0.2301 | 418.1 |
| `tfHuber`-spec.[†] | 0.3239 | 0.2555 | 527.9 | 0.3803 | 0.2669 | 503.8 | 0.3852 | 0.2900 | 515.0 |

[†] Three trials only due to runtime; others rows are 100-trial averages.

Among the 100-trial averages compared in boldface, QTES attains the smallest relative spectral error on Gaussian and Student-$t(5)$, and on lognormal it remains within 0.0075 of WM-oracle while delivering the best subspace recovery. SCM and OAS remain competitive on the Gaussian subspace metric and are substantially faster, while MCD is slower and does not provide a compensating accuracy advantage on this benchmark. The WM-adaptive and MW-Ustat-adaptive diagnostics perform substantially worse than their corresponding oracle rows and than QTES. The MW-Ustat-oracle row is more competitive, but remains less accurate than QTES and WM-oracle on all three clean distributions in relative spectral error. The three-run `tfHuber` checks show that adaptive Huber covariance can be accurate on these clean instances, but the cost is large.

### 3.4 Sample-Size Scaling and Oracle Approximation

This experiment studies how the selector behaves as the sample size grows. We fix the dimension at $d = 200$ and vary the sample size $n \in \{100, 200, 400, 800, 1600, 3200, 5000\}$. The target covariance and data-generating mechanisms remain identical to those in Section 3.1. Each (distribution, $n$, method) setting is averaged over 100 independent trials. WM-oracle is again given the exact clean-distribution scale $\sigma_0$ and serves as the scale-known benchmark within the clipped family.

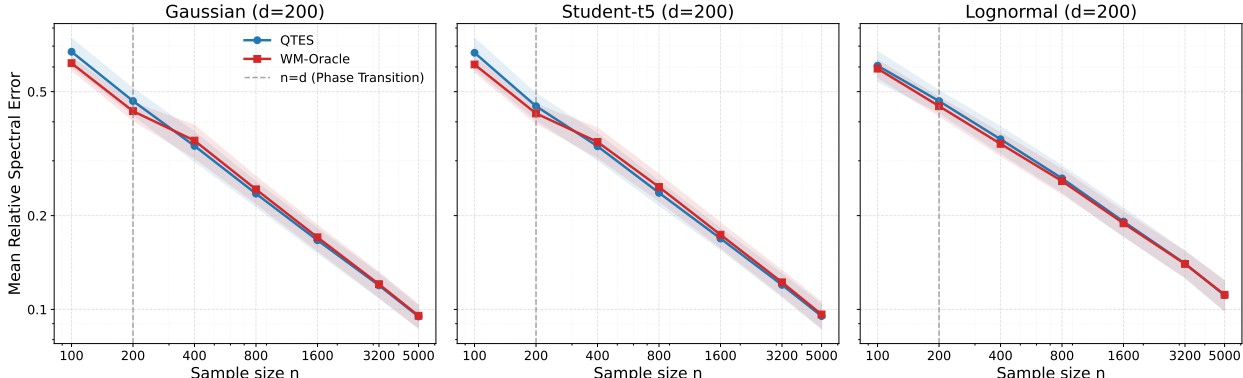

Figure 1: Sample-size scaling of the relative spectral error of QTES and the WM-oracle at fixed dimension $d = 200$. The vertical dashed line indicates the phase transition threshold $n = d$. Shaded regions represent $\pm 1$ standard deviation over 100 independent trials. Both axes are on a logarithmic scale.

Figure 1 shows a clear change around $n = d$. When $n < d$, QTES is more conservative than WM-oracle; once $n$ reaches the dimension, the gap becomes small. When $n < d$, QTES is consistently more conservative than the WM-oracle under all three distributions. At $n = 100$, the mean relative spectral errors are 0.6714 versus 0.6169 for Gaussian data, 0.6664 versus 0.6111 for Student-$t(5)$, and 0.6057 versus 0.5927 for lognormal data. This behavior matches the mathematical cost of the trace relaxation ($\|B\|_{\mathrm{op}} \leq \mathrm{tr}(B)$) used in Lemma 2: when the dimension is large relative to the sample size, the trace inflation makes the QTES score overestimate the clipping bias and therefore encourages conservative under-clipping.

Once $n$ reaches the ambient dimension, however, the gap becomes much smaller. At $n = 400$, QTES slightly outperforms WM-oracle for Gaussian data (0.3348 versus 0.3482) and Student-$t(5)$ data (0.3339 versus 0.3448), while remaining close on lognormal data (0.3517 versus 0.3401). By $n = 5000$, the two methods are very close across all three distributions; for example, on lognormal data the mean errors are 0.1110 for QTES and 0.1114 for WM-oracle.

## 3.5 Empirical Contamination Stress Test

This experiment checks robustness to gross contamination, which is outside the clean i.i.d. setting used in the main theorem. We start from standardized lognormal inliers and replace 5% of the sample by Gaussian outliers drawn from $\mathcal{N}(0, c\Sigma)$ with inflation factor $c \in \{1, 25, 100, 400\}$.

Table 3: Gross-contamination stress test with standardized lognormal inliers at $(n, d) = (400, 200)$. Entries are relative spectral errors.

| Method | Trials | $c = 1$ | $c = 25$ | $c = 100$ | $c = 400$ |
|---|---|---|---|---|---|
| QTES | 100 | 0.3484 | 0.3833 | 0.3911 | 0.3873 |
| SCM | 100 | 0.5867 | 3.5147 | 13.5848 | 56.1519 |
| OAS | 100 | 0.4824 | 3.1133 | 12.7698 | 53.5103 |
| MCD | 100 | 0.4116 | 0.4256 | 0.4270 | 0.4194 |
| WM-oracle | 100 | 0.3406 | 0.3563 | 0.3549 | 0.3547 |
| WM-adaptive | 100 | 0.4462 | 0.4424 | 0.4455 | 0.4460 |
| MW-Ustat-oracle | 100 | 0.3853 | 0.3602 | 0.3619 | 0.3621 |
| MW-Ustat-adaptive | 100 | 0.5278 | 0.5215 | 0.5241 | 0.5244 |
| **tfHuber**-elem.[†] | 3 | 0.3124 | 2.1182 | 7.6333 | 29.6632 |
| **tfHuber**-spec.[†] | 3 | 0.4047 | 1.0931 | 0.9105 | 0.8989 |

[†] Three trials only due to runtime; others rows are 100-trial averages.

Table 3 shows that SCM and OAS deteriorate rapidly as the outlier inflation grows, whereas the robust methods remain much more stable. QTES changes only modestly across the four contamination levels and stays close to the oracle-assisted WM and MW-Ustat references, which are given clean-inlier scale information. The adaptive scale references are more conservative here, reflecting the cost of scale selection in this stress test. The `tfHuber` rows are included only as three-run diagnostics because of runtime; the elementwise mode is strongly affected by large inflation, while the spectrumwise mode is more stable but remains above QTES for the larger inflation factors.

## 4   Discussion and conclusion

We have studied a narrow but important tuning problem: how to choose the clipping level in a radial clipped covariance estimator. QTES combines two data-computable quantities, a variance certificate built from held-out clipped matrices and a scalar estimate of the tail energy removed by clipping, into a variance-plus-bias score. In the simulations, it remains stable under the contamination stress test, and stays close to the scale-oracle clipped benchmark in the regimes considered.

The main theoretical limitation is the size of the selector remainder. The present proof uses a global fourth-moment tail bound and the trace relaxation underlying the tail-energy surrogate, which prevents the bound from matching the sharp direct effective-rank rates available under stronger assumptions. Closing this gap while keeping the rule fully data-driven would likely require a more local analysis of the tail behavior near the best clipping radius.

A separate practical limitation is split variability, especially under contamination. A natural stabilization is to rerun QTES with several independent cross-fitting permutations and aggregate the resulting covariance estimates by an operator-norm medoid, in the spirit of the metric-space median ideas of Hsu & Sabato (2016). Proving a formal amplification guarantee for this repeated-split version is left open, because the repeated outputs are coupled through the common sample.

Another natural extension is to decouple selection from final estimation. One could use QTES to select $\hat{\gamma}_{\mathrm{QTES}}$, recompute a single clipping radius from all $n$ observations, and then refit the covariance on the full clipped sample. This variant is algorithmically simple, but its analysis would require new arguments because the same sample is reused for both calibration and estimation.

In summary, QTES is a practical calibration rule for radial clipping: it is simple to implement, does not require a user-supplied scale range, and comes with a finite-sample selector guarantee under a finite fourth-moment assumption.

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

# A    Proofs of the Main Variance and Bias Ingredients

This section proves the two basic ingredients used in the main result: the variance certificate and the tail-energy bias bound. As in Section 2, for each fold $k$ and grid point $\gamma$, let $Z_k^{\text{test}} \sim P$ be independent of $\mathcal{F}_k$, and write
$$Y_k^{\text{test}}(\gamma) := (\|Z_k^{\text{test}}\|_2^2 - \tau_k(\gamma)^2)_+.$$
Conditional on $\mathcal{F}_k$, this is the same law as any held-out variable $Y_i^{(k)}(\gamma)$ with $i \in I_k$.

## A.1    Proof of Theorem 1

*Proof.* Fix $\gamma \in \mathcal{G}$. For fold $k \in \{1, 2\}$, condition on $\mathcal{F}_k := \sigma(\Pi, \{Z_i\}_{i \in J_k})$. If $\tau_k(\gamma) = 0$, then $A_i^{(k)}(\gamma) = 0$, $V_k^\star(\gamma) = 0$, and $\Psi_k^{\text{raw}}(\gamma) = 0$. Otherwise, the matrices $\{A_i^{(k)}(\gamma)\}_{i \in I_k}$ are conditionally i.i.d. self-adjoint with eigenvalues in $[0, 1]$. For any fixed $i \in I_k$, define

$$M_k(\gamma) := \mathbb{E}_P[A_i^{(k)}(\gamma) \mid \mathcal{F}_k], \qquad \hat{M}_k(\gamma) := \frac{1}{n_k} \sum_{i \in I_k} A_i^{(k)}(\gamma).$$

Then $\hat{\Sigma}_k^{\text{raw}}(\gamma) = \tau_k(\gamma)^2 \, \hat{M}_k(\gamma)$ and $\mathbb{E}_P[\hat{\Sigma}_k^{\text{raw}}(\gamma) \mid \mathcal{F}_k] = \tau_k(\gamma)^2 \, M_k(\gamma)$. Applying Wang & Ramdas (2025, Theorem 3.1) to $\{A_i^{(k)}(\gamma)\}_{i \in I_k}$ yields the one-sided deviation bound with radius $D_{n_k, d}(\alpha_\gamma; V_k^\star(\gamma))$. Applying the same inequality to $\{I_d - A_i^{(k)}(\gamma)\}_{i \in I_k}$ gives the lower tail. A union bound over both tails, both folds, and all $|\mathcal{G}|$ points guarantees the envelope with probability at least $1 - \delta_{\text{var}}$. Summing with weights $n_k/n_{\text{ret}}$ proves (7). The suffix maximum inequality is deterministic. $\qquad\square$

## A.2    Proof of Lemma 2

*Proof.* Fix $k$ and $\gamma$, and define

$$\alpha(z) := \begin{cases} \min\{1, \tau_k(\gamma)/\|z\|_2\}, & \|z\|_2 > 0, \\ 1, & \|z\|_2 = 0. \end{cases}$$

Then $\alpha(z)z$ is exactly the clipped vector appearing in (3). Since $Z_k^{\text{test}}$ is independent of $\mathcal{F}_k$ and distributed as $P$,
$$\mathbb{E}_P[\hat{\Sigma}_k^{\text{raw}}(\gamma) \mid \mathcal{F}_k] = \mathbb{E}_P\big[\alpha(Z_k^{\text{test}})^2 Z_k^{\text{test}}(Z_k^{\text{test}})^\top \mid \mathcal{F}_k\big].$$

Hence

$$B_k(\gamma) = \mathbb{E}_P\big[(1 - \alpha(Z_k^{\text{test}})^2)Z_k^{\text{test}}(Z_k^{\text{test}})^\top \mid \mathcal{F}_k\big].$$

Since $0 \le 1 - \alpha(Z_k^{\text{test}})^2 \le 1$, the integrand is positive semidefinite, so $B_k(\gamma) \succeq 0$. Therefore $\|B_k(\gamma)\|_{\text{op}} \le \text{tr}(B_k(\gamma))$. Moreover,

$$\text{tr}(B_k(\gamma)) = \mathbb{E}_P\big[(1 - \alpha(Z_k^{\text{test}})^2)\|Z_k^{\text{test}}\|_2^2 \mid \mathcal{F}_k\big].$$

If $\|Z_k^{\text{test}}\|_2 \le \tau_k(\gamma)$, then $\alpha(Z_k^{\text{test}}) = 1$ and the contribution is zero. If $\|Z_k^{\text{test}}\|_2 > \tau_k(\gamma)$, then $\alpha(Z_k^{\text{test}})^2 = \tau_k(\gamma)^2/\|Z_k^{\text{test}}\|_2^2$, so

$$(1 - \alpha(Z_k^{\text{test}})^2)\|Z_k^{\text{test}}\|_2^2 = \|Z_k^{\text{test}}\|_2^2 - \tau_k(\gamma)^2.$$

Thus

$$\text{tr}(B_k(\gamma)) = \mathbb{E}_P\big[Y_k^{\text{test}}(\gamma) \mid \mathcal{F}_k\big].$$

$\square$

## B  Auxiliary Results and Proofs

This section turns the two ingredients into the main selector guarantee. We first state a generic result: any valid upper bound on the clipping bias can be combined with the variance certificate. We then prove that the QTES block-quantile bias estimate has the required calibration.

### B.1  A Generic Oracle Template

We begin with a generic template. Its role is to separate variance control from bias calibration: once a simultaneous upper bound for the bias term is available, the statement below converts it into a uniform risk bound over the entire clipping grid.

**Theorem 5** (Generic oracle inequality from a valid bias upper bound). *Suppose a data-dependent quantity* $\widehat{b}_k^{\text{UCB}}(\gamma)$ *is constructed from* $\{Y_i^{(k)}(\gamma) : i \in I_k\}$ *such that with probability at least* $1 - \delta_{\text{bias}}$,

$$\mathbb{E}_P\big[Y_k^{\text{test}}(\gamma) \mid \mathcal{F}_k\big] \le \widehat{b}_k^{\text{UCB}}(\gamma) \qquad \text{simultaneously for all } k \in \{1, 2\}, \ \gamma \in \mathcal{G}. \tag{16}$$

*Define*

$$\widehat{U}_{\text{BiasUCB}}(\gamma) := \overline{\Psi}^{\text{raw}}(\gamma) + \sum_{k=1}^{2} \frac{n_k}{n_{\text{ret}}} \widehat{b}_k^{\text{UCB}}(\gamma). \tag{17}$$

*If the event* (7) *of Theorem 1 also holds, then with probability at least* $1 - \delta_{\text{var}} - \delta_{\text{bias}}$, *simultaneously for all* $\gamma \in \mathcal{G}$,

$$\left\|\widehat{\Sigma}^{\text{raw}}(\gamma) - \Sigma\right\|_{\text{op}} \le \widehat{U}_{\text{BiasUCB}}(\gamma). \tag{18}$$

*Consequently, for* $\hat{\gamma}_{\text{BiasUCB}} \in \arg\min_{\gamma \in \mathcal{G}} \widehat{U}_{\text{BiasUCB}}(\gamma)$,

$$\left\|\widehat{\Sigma}^{\text{raw}}(\hat{\gamma}_{\text{BiasUCB}}) - \Sigma\right\|_{\text{op}} \le \min_{\gamma \in \mathcal{G}} \widehat{U}_{\text{BiasUCB}}(\gamma). \tag{19}$$

*Proof.* On the intersection of the events (7) and (16), for every $\gamma \in \mathcal{G}$,

$$\left\|\widehat{\Sigma}^{\text{raw}}(\gamma) - \Sigma\right\|_{\text{op}} \le \left\|\widehat{\Sigma}^{\text{raw}}(\gamma) - \bar{\Sigma}_P^{\text{raw}}(\gamma)\right\|_{\text{op}} + \left\|\bar{\Sigma}_P^{\text{raw}}(\gamma) - \Sigma\right\|_{\text{op}}.$$

The first term is bounded by $\overline{\Psi}^{\text{raw}}(\gamma)$ from Theorem 1. The second term is bounded by $b(\gamma)$ from Lemma 2, which is in turn bounded by $\sum_{k=1}^{2}(n_k/n_{\text{ret}})\widehat{b}_k^{\text{UCB}}(\gamma)$ on the event (16). This proves (18). Taking the minimum over $\gamma$ proves (19). $\square$

## B.2 Calibration of the QTES Bias Score

The next theorem checks the condition required by the generic template. It shows that the block-quantile estimate of the tail-energy term is uniformly close to its conditional mean over all folds and grid points.

**Theorem 6** (Uniform calibration of the QTES bias score). *Assume* (1) *and* (11)*, and define*

$$s_\star := \left\lfloor \frac{n_1}{m_{\text{blocks}}} \right\rfloor.$$

*For each fold $k$ and grid point $\gamma$, let*

$$\mu_k(\gamma) := \mathbb{E}_P\big[Y_k^{\text{test}}(\gamma) \mid \mathcal{F}_k\big], \qquad v_k(\gamma) := \text{Var}_P\big(Y_k^{\text{test}}(\gamma) \mid \mathcal{F}_k\big).$$

*Then with probability at least $1 - \delta_{\text{bias}}$, simultaneously for all $k \in \{1,2\}$ and $\gamma \in \mathcal{G}$,*

$$\big|\widehat{b}_k^{\text{QTES}}(\gamma) - \mu_k(\gamma)\big| \leq 10\sqrt{\frac{v_k(\gamma)}{s_\star}}. \tag{20}$$

*In particular, since $Y_i^{(k)}(\gamma) \leq \|Z_i\|_2^2$, the same event implies*

$$\big|\widehat{b}_k^{\text{QTES}}(\gamma) - \mu_k(\gamma)\big| \leq \Delta_n, \qquad \Delta_n := 10\sqrt{\frac{\mathbb{E}_P\|Z_1\|_2^4}{s_\star}}. \tag{21}$$

*Consequently, with the same probability,*

$$\big|\widehat{b}^{\text{QTES}}(\gamma) - b(\gamma)\big| \leq \Delta_n \qquad \text{simultaneously for all } \gamma \in \mathcal{G}. \tag{22}$$

*Proof.* Fix a fold $k \in \{1,2\}$ and a grid point $\gamma \in \mathcal{G}$. Let $\Pi_{k,\gamma}$ denote the independent random partition of $I_k$ into $m := m_{\text{blocks}}$ blocks whose sizes differ by at most one, and define

$$\mathcal{H}_{k,\gamma} := \sigma(\mathcal{F}_k, \Pi_{k,\gamma}).$$

Write the blocks as $G_1, \ldots, G_m$, let $s_j := |G_j|$, and note that $s_j \geq s_\star = \lfloor n_1/m \rfloor$ for all $j$. Define the block means

$$U_j^{(k)}(\gamma) := \frac{1}{s_j} \sum_{i \in G_j} Y_i^{(k)}(\gamma), \qquad j = 1, \ldots, m.$$

Conditionally on $\mathcal{H}_{k,\gamma}$, the variables $U_1^{(k)}(\gamma), \ldots, U_m^{(k)}(\gamma)$ are independent and each has mean $\mu_k(\gamma)$. Moreover,

$$\text{Var}_P\big(U_j^{(k)}(\gamma) \mid \mathcal{H}_{k,\gamma}\big) \leq \frac{v_k(\gamma)}{s_j} \leq \frac{v_k(\gamma)}{s_\star}.$$

Set

$$t_k(\gamma) := 10\sqrt{\frac{v_k(\gamma)}{s_\star}}.$$

By Chebyshev's inequality, for every $j$,

$$\mathbb{P}\Big(U_j^{(k)}(\gamma) > \mu_k(\gamma) + t_k(\gamma) \mid \mathcal{H}_{k,\gamma}\Big) \leq 10^{-2},$$

and similarly

$$\mathbb{P}\Big(U_j^{(k)}(\gamma) < \mu_k(\gamma) - t_k(\gamma) \mid \mathcal{H}_{k,\gamma}\Big) \leq 10^{-2}.$$

Let $U_{(1)}^{(k)}(\gamma) \leq \cdots \leq U_{(m)}^{(k)}(\gamma)$ denote the order statistics, and let

$$r := \lceil 0.85m \rceil, \qquad \widehat{b}_k^{\text{QTES}}(\gamma) = U_{(r)}^{(k)}(\gamma).$$

If $\widehat{b}_k^{\mathrm{QTES}}(\gamma) > \mu_k(\gamma) + t_k(\gamma)$, then at least $m - r + 1 \geq 0.15m$ block means exceed $\mu_k(\gamma) + t_k(\gamma)$. Hence, with

$$N_+ := \sum_{j=1}^m \mathbf{1}\Big\{U_j^{(k)}(\gamma) > \mu_k(\gamma) + t_k(\gamma)\Big\},$$

we have, since $N_+$ is a sum of independent Bernoulli variables with conditional success probabilities at most $0.01$,

$$\mathbb{P}\Big(\widehat{b}_k^{\mathrm{QTES}}(\gamma) > \mu_k(\gamma) + t_k(\gamma) \mid \mathcal{H}_{k,\gamma}\Big) \leq \mathbb{P}\big(N_+ \geq 0.15m \mid \mathcal{H}_{k,\gamma}\big) \leq \exp\big(-mD(0.15\|0.01)\big),$$

where $D(a\|b) := a\log(a/b) + (1-a)\log((1-a)/(1-b))$ is the Bernoulli relative entropy. Likewise, if $\widehat{b}_k^{\mathrm{QTES}}(\gamma) < \mu_k(\gamma) - t_k(\gamma)$, then at least $r \geq 0.85m$ block means fall below $\mu_k(\gamma) - t_k(\gamma)$, so with

$$N_- := \sum_{j=1}^m \mathbf{1}\Big\{U_j^{(k)}(\gamma) < \mu_k(\gamma) - t_k(\gamma)\Big\},$$

we obtain by the same Poisson-binomial Chernoff bound

$$\mathbb{P}\Big(\widehat{b}_k^{\mathrm{QTES}}(\gamma) < \mu_k(\gamma) - t_k(\gamma) \mid \mathcal{H}_{k,\gamma}\Big) \leq \mathbb{P}\big(N_- \geq 0.85m \mid \mathcal{H}_{k,\gamma}\big) \leq \exp\big(-mD(0.85\|0.01)\big).$$

Numerically, $D(0.15\|0.01) \approx 0.2766$ and $D(0.85\|0.01) > 0.2766$. Therefore, for each pair $(k,\gamma)$,

$$\mathbb{P}\Big(\big|\widehat{b}_k^{\mathrm{QTES}}(\gamma) - \mu_k(\gamma)\big| > t_k(\gamma)\Big) \leq 2\exp(-0.2766\,m).$$

By definition, $m = m_{\mathrm{base}} \geq 4\log(4|\mathcal{G}|/\delta_{\mathrm{bias}})$. A union bound over the $2|\mathcal{G}|$ choices of $(k,\gamma)$ yields total failure probability at most

$$4|\mathcal{G}|\exp(-0.2766\,m) \leq 4|\mathcal{G}| \left(\frac{\delta_{\mathrm{bias}}}{4|\mathcal{G}|}\right)^{4\cdot 0.2766} \leq \delta_{\mathrm{bias}},$$

because $4 \cdot 0.2766 > 1$ and $\delta_{\mathrm{bias}}/(4|\mathcal{G}|) \leq 1$. This proves (20).

For (21), note that $0 \leq Y_i^{(k)}(\gamma) \leq \|Z_i\|_2^2$, so

$$v_k(\gamma) \leq \mathbb{E}_P\big[(Y_i^{(k)}(\gamma))^2 \mid \mathcal{F}_k\big] \leq \mathbb{E}_P\|Z_1\|_2^4.$$

Finally,

$$\big|\widehat{b}^{\mathrm{QTES}}(\gamma) - b(\gamma)\big| = \left|\sum_{k=1}^2 \frac{n_k}{n_{\mathrm{ret}}}\big(\widehat{b}_k^{\mathrm{QTES}}(\gamma) - \mu_k(\gamma)\big)\right| \leq \sum_{k=1}^2 \frac{n_k}{n_{\mathrm{ret}}}\big|\widehat{b}_k^{\mathrm{QTES}}(\gamma) - \mu_k(\gamma)\big| \leq \Delta_n,$$

which proves (22). $\qquad\square$

### B.3    Consequences for the QTES Score

Applying Theorem 5 with the calibration supplied by Theorem 6 shows that the QTES score is, up to an additive constant independent of $\gamma$, a strict simultaneous upper confidence family over $\mathcal{G}$. We record this consequence separately because it is the form used in the proof of Theorem 3.

**Corollary 7** (Simultaneous UCB form)**.** *With probability at least $1 - \delta$, the following statements hold simultaneously for all $\gamma \in \mathcal{G}$:*

$$\Big\|\widehat{\Sigma}^{\mathrm{raw}}(\gamma) - \Sigma\Big\|_{\mathrm{op}} \leq \widehat{S}_{\mathrm{QTES}}(\gamma) + \Delta_n. \tag{23}$$

*Hence the family*

$$\widetilde{U}_{\mathrm{QTES}}(\gamma) := \widehat{S}_{\mathrm{QTES}}(\gamma) + \Delta_n$$

*is a simultaneous upper confidence family over $\mathcal{G}$. Because $\Delta_n$ is independent of $\gamma$, the argmin sets coincide:*

$$\arg\min_{\gamma\in\mathcal{G}} \widehat{S}_{\mathrm{QTES}}(\gamma) = \arg\min_{\gamma\in\mathcal{G}} \widetilde{U}_{\mathrm{QTES}}(\gamma). \tag{24}$$

*In particular, $\hat{\gamma}_{\mathrm{QTES}}$ exactly minimizes a strict simultaneous upper confidence family.*

*Proof.* Fix $\gamma \in \mathcal{G}$ and work on the intersection of the events from Theorem 1 and Theorem 6. By the triangle inequality,

$$\left\|\hat{\Sigma}^{\text{raw}}(\gamma) - \Sigma\right\|_{\text{op}} \leq \left\|\hat{\Sigma}^{\text{raw}}(\gamma) - \bar{\Sigma}_P^{\text{raw}}(\gamma)\right\|_{\text{op}} + \left\|\bar{\Sigma}_P^{\text{raw}}(\gamma) - \Sigma\right\|_{\text{op}}.$$

The first term is bounded by $\overline{\Psi}^{\text{raw}}(\gamma)$ from Theorem 1. The second term is bounded by $b(\gamma)$ from Lemma 2. By (22),

$$b(\gamma) \leq \hat{b}^{\text{QTES}}(\gamma) + \Delta_n.$$

Combining these inequalities gives (23). Since adding the same constant $\Delta_n$ to every score does not change the minimizers, (24) follows immediately. □

Because the additive slack in Corollary 7 does not depend on $\gamma$, minimizing the QTES score is equivalent to minimizing the associated upper confidence family. The next corollary turns this observation into the selected-estimator guarantee stated in Theorem 3.

**Corollary 8** (Near-oracle inequality for the selected QTES estimator). *Under the same event as Corollary 7,*

$$\left\|\hat{\Sigma}^{\text{raw}}(\hat{\gamma}_{\text{QTES}}) - \Sigma\right\|_{\text{op}} \leq \min_{\gamma \in \mathcal{G}} \left\{\overline{\Psi}^{\text{raw}}(\gamma) + b(\gamma)\right\} + 2\Delta_n. \tag{25}$$

*Proof.* Define the ideal observable-plus-bias curve

$$O(\gamma) := \overline{\Psi}^{\text{raw}}(\gamma) + b(\gamma), \qquad \gamma \in \mathcal{G}.$$

On the event of Corollary 7, (22) implies

$$\left|\hat{S}_{\text{QTES}}(\gamma) - O(\gamma)\right| = \left|\hat{b}^{\text{QTES}}(\gamma) - b(\gamma)\right| \leq \Delta_n \qquad \text{for all } \gamma \in \mathcal{G}.$$

Let $\gamma_\star \in \arg\min_{\gamma \in \mathcal{G}} O(\gamma)$. Because $\hat{\gamma}_{\text{QTES}}$ minimizes $\hat{S}_{\text{QTES}}$, we have

$$\hat{S}_{\text{QTES}}(\hat{\gamma}_{\text{QTES}}) \leq \hat{S}_{\text{QTES}}(\gamma_\star) \leq O(\gamma_\star) + \Delta_n.$$

Applying (23) at $\hat{\gamma}_{\text{QTES}}$ yields

$$\left\|\hat{\Sigma}^{\text{raw}}(\hat{\gamma}_{\text{QTES}}) - \Sigma\right\|_{\text{op}} \leq \hat{S}_{\text{QTES}}(\hat{\gamma}_{\text{QTES}}) + \Delta_n \leq O(\gamma_\star) + 2\Delta_n,$$

which is exactly (25). □

## C   Proofs of the Remaining Main-Text Results

This section finishes the remaining main-text proofs by combining the variance and bias events established above.

### C.1   Proof of Theorem 3

*Proof.* Work on the intersection of the events from Theorem 1 and Theorem 6. On this event, the aggregate calibration bound (22) holds for all $\gamma \in \mathcal{G}$, and Corollary 7 yields the simultaneous upper bound (23) for all $\gamma \in \mathcal{G}$. These are exactly the two claims in (13). The selected-estimator inequality (14) is Corollary 8. Since $\delta_{\text{var}} = \delta_{\text{bias}} = \delta/2$, the intersection event has probability at least $1 - \delta$. □

### C.2   Proof of Corollary 4

*Proof.* Define

$$O(\gamma) := \overline{\Psi}^{\text{raw}}(\gamma) + b(\gamma), \qquad \gamma \in \mathcal{G},$$

and let $\gamma_\star \in \arg\min_{\gamma \in \mathcal{G}} O(\gamma)$. On the event of Corollary 7, the calibration bound (22) gives

$$\left|\widehat{S}_{\mathrm{QTES}}(\gamma) - O(\gamma)\right| \leq \Delta_n \qquad \text{for all } \gamma \in \mathcal{G}.$$

Because $\hat{\gamma}_{\mathrm{QTES}}$ minimizes $\widehat{S}_{\mathrm{QTES}}$,

$$O(\hat{\gamma}_{\mathrm{QTES}}) \leq \widehat{S}_{\mathrm{QTES}}(\hat{\gamma}_{\mathrm{QTES}}) + \Delta_n \leq \widehat{S}_{\mathrm{QTES}}(\gamma_\star) + \Delta_n \leq O(\gamma_\star) + 2\Delta_n,$$

which proves (15). The $\Gamma_\varepsilon$ claim is immediate when $\varepsilon \geq 2\Delta_n$. If the minimizer $\gamma_\star$ is unique and every other grid point lies more than $2\Delta_n$ above $O(\gamma_\star)$, then (15) forces $\hat{\gamma}_{\mathrm{QTES}} = \gamma_\star$. □

## D  Extension to an Unknown Mean

The main theorem assumes centered data. This section explains how to use the same selection idea when the mean is unknown. The procedure is simple: estimate the mean on the training fold $J_k$, subtract this estimate from the held-out fold $I_k$, and then run the same QTES score. The theorem below keeps the mean-estimation error explicit. A poor centering estimator does not invalidate the score construction, but it adds a squared-error term to the final covariance bound.

### D.1  Mathematical effect of foldwise centering

In this appendix only, let $X_i \sim P$ have mean $\mu$ and covariance

$$\Sigma = \mathbb{E}_P[(X_1 - \mu)(X_1 - \mu)^\top].$$

Write $W_i := X_i - \mu$. For each fold $k$, let $\hat{\mu}^{(k)}$ be any centering estimator computed only from the training fold $J_k$, and set

$$\Delta_k := \hat{\mu}^{(k)} - \mu, \qquad V_i^{(k)} := X_i - \hat{\mu}^{(k)} = W_i - \Delta_k.$$

Conditional on $\mathcal{F}_k := \sigma(\Pi, \{X_i : i \in J_k\})$, the vector $\Delta_k$ and the clipping radius $\tau_k(\gamma)$ are fixed. The unknown-mean version of QTES is obtained by replacing every centered vector $Z_i$ in Section 2 by $V_i^{(k)}$ on fold $k$.

**Proposition 9** (Tail-energy control with estimated centering). *Fix a fold $k$ and a grid point $\gamma$. Let*

$$\alpha(v) := \begin{cases} \min\{1, \tau_k(\gamma)/\|v\|_2\}, & \|v\|_2 > 0, \\ 1, & \|v\|_2 = 0, \end{cases}$$

*and let $V^{(k)} := X - \hat{\mu}^{(k)}$, where $X \sim P$ is independent of $\mathcal{F}_k$. Define*

$$B_k^\mu(\gamma) := \Sigma - \mathbb{E}_P\left[\alpha(V^{(k)})^2 V^{(k)}(V^{(k)})^\top \mid \mathcal{F}_k\right].$$

*Then*

$$B_k^\mu(\gamma) = \mathbb{E}_P\left[(1 - \alpha(V^{(k)})^2)V^{(k)}(V^{(k)})^\top \mid \mathcal{F}_k\right] - \Delta_k \Delta_k^\top \tag{26}$$

*and*

$$\|B_k^\mu(\gamma)\|_{\mathrm{op}} \leq \mathbb{E}_P\left[(\|V^{(k)}\|_2^2 - \tau_k(\gamma)^2)_+ \mid \mathcal{F}_k\right] + \|\Delta_k\|_2^2. \tag{27}$$

*Proof.* Since $V^{(k)} = X - \hat{\mu}^{(k)} = W - \Delta_k$ and $\mathbb{E}_P[V^{(k)} \mid \mathcal{F}_k] = -\Delta_k$,

$$\Sigma = \mathbb{E}_P[WW^\top] = \mathbb{E}_P[(V^{(k)} + \Delta_k)(V^{(k)} + \Delta_k)^\top \mid \mathcal{F}_k] = \mathbb{E}_P[V^{(k)}(V^{(k)})^\top \mid \mathcal{F}_k] - \Delta_k \Delta_k^\top.$$

Subtracting the conditional mean of the clipped covariance gives (26). For any unit vector $u$,

$$u^\top B_k^\mu(\gamma)u = \mathbb{E}_P[(1 - \alpha(V^{(k)})^2)(u^\top V^{(k)})^2 \mid \mathcal{F}_k] - (u^\top \Delta_k)^2.$$

The first term is nonnegative and bounded by the trace of its matrix, while the second is bounded by $\|\Delta_k\|_2^2$. Hence

$$|u^\top B_k^\mu(\gamma)u| \leq \mathbb{E}_P[(1 - \alpha(V^{(k)})^2)\|V^{(k)}\|_2^2 \mid \mathcal{F}_k] + \|\Delta_k\|_2^2.$$

Taking the supremum over $\|u\|_2 = 1$ and using the definition of $\alpha$ yields (27). □

The proposition shows that the PSD identity from Lemma 2 is not literally preserved: $B_k^\mu(\gamma)$ may have a negative direction because of $-\Delta_k \Delta_k^\top$. The loss is controlled by $\|\Delta_k\|_2^2$, which is independent of $\gamma$. This is the precise sense in which the centering error is grid-invariant. It raises the theoretical upper envelope, but it does not create another clipping-level bias–variance tradeoff.

We now state the corresponding probabilistic form. Define the shifted held-out tail variables

$$Y_{i,\mu}^{(k)}(\gamma) := (\|V_i^{(k)}\|_2^2 - \tau_k(\gamma)^2)_+, \qquad i \in I_k,$$

construct $\widehat{b}_\mu^{\mathrm{QTES}}(\gamma)$ from these variables by the same block-quantile rule as in (12), and define $\overline{\Psi}_\mu^{\mathrm{raw}}(\gamma)$ from the clipped shifted vectors $V_i^{(k)}$. Let

$$b_\mu(\gamma) := \sum_{k=1}^2 \frac{n_k}{n_{\mathrm{ret}}} \mathbb{E}_P\left[(\|V^{(k)}\|_2^2 - \tau_k(\gamma)^2)_+ \mid \mathcal{F}_k\right], \qquad \eta_\mu := \sum_{k=1}^2 \frac{n_k}{n_{\mathrm{ret}}} \|\Delta_k\|_2^2.$$

**Corollary 10** (Unknown-mean QTES bound). *Assume the shifted fourth moment is finite,*

$$M_{4,\mu} := \max_{k \in \{1,2\}} \left(\mathbb{E}_P[\|W - \Delta_k\|_2^4 \mid \mathcal{F}_k]\right)^{1/4} < \infty \qquad \text{almost surely,}$$

*and assume the block-feasibility condition* (11). *Define*

$$\Delta_{n,\mu} := 10 M_{4,\mu}^2 \, s_\star^{-1/2}, \qquad \widehat{S}_{\mathrm{QTES}}^\mu(\gamma) := \overline{\Psi}_\mu^{\mathrm{raw}}(\gamma) + \widehat{b}_\mu^{\mathrm{QTES}}(\gamma).$$

*Then, with probability at least $1 - \delta$, the following inequalities hold simultaneously for all $\gamma \in \mathcal{G}$:*

$$\left\|\widehat{\Sigma}_\mu^{\mathrm{raw}}(\gamma) - \Sigma\right\|_{\mathrm{op}} \leq \widehat{S}_{\mathrm{QTES}}^\mu(\gamma) + \Delta_{n,\mu} + \eta_\mu, \qquad \gamma \in \mathcal{G}. \tag{28}$$

*Consequently, for*

$$\hat{\gamma}_{\mathrm{QTES}}^\mu \in \arg\min_{\gamma \in \mathcal{G}} \widehat{S}_{\mathrm{QTES}}^\mu(\gamma),$$

*one obtains the selected-estimator bound*

$$\left\|\widehat{\Sigma}_\mu^{\mathrm{raw}}(\hat{\gamma}_{\mathrm{QTES}}^\mu) - \Sigma\right\|_{\mathrm{op}} \leq \min_{\gamma \in \mathcal{G}} \left\{\overline{\Psi}_\mu^{\mathrm{raw}}(\gamma) + b_\mu(\gamma)\right\} + 2\Delta_{n,\mu} + \eta_\mu. \tag{29}$$

*Proof.* The proof uses the same foldwise conditioning as the centered argument. For each fold $k$, condition on $\mathcal{F}_k$, including the foldwise centering estimate $\hat{\mu}^{(k)}$. Under this conditioning, $\Delta_k$ and $\tau_k(\gamma)$ are fixed for every $\gamma$, and the shifted held-out vectors $\{V_i^{(k)} : i \in I_k\}$ are independent. The variance-certificate proof of Theorem 1 applies verbatim to the normalized clipped matrices formed from $V_i^{(k)}$. The block-quantile calibration proof of Theorem 6 also applies verbatim to $Y_{i,\mu}^{(k)}(\gamma)$, with the conditional fourth-moment bound $M_{4,\mu}^4$ replacing $\mathbb{E}_P \|Z_1\|_2^4$. Integrating these foldwise conditional bounds and taking the same union bounds over folds, tails, and grid points as in the centered proof gives an event of probability at least $1 - \delta$ on which, simultaneously over $\gamma \in \mathcal{G}$,

$$\left\|\widehat{\Sigma}_\mu^{\mathrm{raw}}(\gamma) - \bar{\Sigma}_{P,\mu}^{\mathrm{raw}}(\gamma)\right\|_{\mathrm{op}} \leq \overline{\Psi}_\mu^{\mathrm{raw}}(\gamma), \qquad \left|\widehat{b}_\mu^{\mathrm{QTES}}(\gamma) - b_\mu(\gamma)\right| \leq \Delta_{n,\mu},$$

where $\bar{\Sigma}_{P,\mu}^{\mathrm{raw}}(\gamma)$ is the fold-conditional mean of the unknown-mean clipped estimator. Proposition 9, summed over folds, gives

$$\left\|\Sigma - \bar{\Sigma}_{P,\mu}^{\mathrm{raw}}(\gamma)\right\|_{\mathrm{op}} \leq b_\mu(\gamma) + \eta_\mu.$$

Combining the last three displays proves (28). The selected-estimator inequality follows exactly as in Corollary 8: compare $\widehat{S}_{\mathrm{QTES}}^\mu$ with $\overline{\Psi}_\mu^{\mathrm{raw}} + b_\mu$ uniformly, and use the fact that $\hat{\gamma}_{\mathrm{QTES}}^\mu$ minimizes $\widehat{S}_{\mathrm{QTES}}^\mu$. $\qquad\square$

### D.2 Two examples

The previous bound is deliberately agnostic about the mean estimator. It is valid for any foldwise centering vector, but the statistical quality of the final covariance estimator depends on the size of $\eta_\mu$. We record two standard choices.

**Sample mean.**   Under clean sampling, the empirical mean on $J_k$ satisfies the elementary Chebyshev bound

$$\mathbb{P}\left(\|\hat{\mu}_{\mathrm{SM}}^{(k)} - \mu\|_2^2 \geq \frac{\mathrm{tr}(\Sigma)}{|J_k|\delta_\mu}\right) \leq \delta_\mu. \tag{30}$$

Thus the sample mean is adequate at fixed confidence in a clean finite-variance model, and the centering price is of order $\mathrm{tr}(\Sigma)/|J_k|$ up to the confidence factor. Its drawback is the polynomial $1/\delta_\mu$ dependence and its sensitivity to outliers.

**Geometric median-of-means.**   A robust alternative is the geometric median-of-means (MOM) estimator: split $J_k$ into blocks, compute the mean within each block, and return the spatial median of the block means. Corollary 4.1 of Minsker (2015) gives, for a suitable logarithmic number of blocks and finite second moment,

$$\|\hat{\mu}_{\mathrm{MOM}}^{(k)} - \mu\|_2^2 \leq C\,\frac{\mathrm{tr}(\Sigma)\log(C'/\delta_\mu)}{|J_k|} \tag{31}$$

with probability at least $1 - \delta_\mu$, for numerical constants $C, C' > 0$. A union bound over the two folds gives

$$\eta_\mu = \mathcal{O}\left(\frac{\mathrm{tr}(\Sigma)\log(1/\delta_\mu)}{\min_k |J_k|}\right)$$

with high probability. Since $|J_k|$ is of order $n$, this is typically smaller than the global fourth-moment selector slack in Theorem 3; for example, under the usual $L_4$–$L_2$ comparison it is of order $\mathrm{tr}(\Sigma)\log(1/\delta_\mu)/n$, whereas the selector slack is of order $\mathrm{tr}(\Sigma)\sqrt{\log(|\mathcal{G}|/\delta)/n}$. The unknown mean therefore adds a standard robust-mean estimation price, but it does not create a new clipping-level calibration problem.

The slack $\Delta_{n,\mu}$ is also stable under accurate centering. If $M_{4,W} := (\mathbb{E}_P\|W_1\|_2^4)^{1/4}$ and $\max_k \|\Delta_k\|_2 \leq r_\mu$, then

$$\mathbb{E}_P[\|W - \Delta_k\|_2^4 \mid \mathcal{F}_k] \leq 8\{M_{4,W}^4 + r_\mu^4\}.$$

Thus $\Delta_{n,\mu}$ has the same order as the original $\Delta_n$ whenever the mean estimator is consistent at the scale relevant to the problem.

### D.3 Empirical check with a nonzero mean

To make this point concrete, we perform a nonzero-mean check by shifting every observation by the vector $\mathbf{1}_d$. The algorithms in the Sample Mean and MOM rows are not given this vector; they estimate the mean only from the corresponding training fold before applying QTES to the held-out fold. The Known Mean Reference row subtracts the true vector $\mathbf{1}_d$ and is included only as a reference. The covariance target remains the unshifted covariance $\Sigma$. For reproducibility, the MOM row uses 10 blocks in the clean runs, a logarithmic confidence-boosting scale for the training-fold size $|J_k| = 200$. In the contaminated stress runs, 5% of the observations are replaced by Gaussian outliers drawn from $\mathcal{N}(0, 100\Sigma)$ before the common shift by $\mathbf{1}_d$ is applied. The MOM row uses 30 blocks in these stress runs, equal to $\lceil 3\varepsilon|J_k|\rceil$ with contamination fraction $\varepsilon = 0.05$; this is an empirical robustness choice for the stress test, not an assumption in the clean theory.

Table 4 shows the resulting averages over 100 trials at $(n, d) = (400, 200)$. The clean settings are reported first, followed by the contaminated stress tests. Within each scenario we list the two practical centering rules first, and the known-mean reference last. In the clean settings, estimating the mean changes the relative spectral error only mildly. In the contaminated settings, the ordinary sample mean can be pulled by the inflated observations, which is exactly reflected by the larger errors. The geometric median-of-means centering step largely removes this deterioration and stays close to the known-mean reference. These contamination rows are

Table 4: QTES with true mean $\mu = \mathbf{1}_d$ unknown to the adaptive procedures, over 100 independent trials at $(n, d) = (400, 200)$. Entries report mean $\pm$ standard deviation.

| Experimental Scenario | Mean Estimator | RelSpecErr | SubspaceErr |
|---|---|---|---|
| Gaussian (clean) | Sample mean | $0.34991 \pm 0.03777$ | $0.26875 \pm 0.01378$ |
| | MOM estimator | $0.35049 \pm 0.03770$ | $0.26888 \pm 0.01378$ |
| | Known mean reference | $0.33300 \pm 0.03177$ | $0.26251 \pm 0.01073$ |
| Student-$t(5)$ (clean) | Sample mean | $0.34912 \pm 0.03770$ | $0.26882 \pm 0.01254$ |
| | MOM estimator | $0.34889 \pm 0.03785$ | $0.26877 \pm 0.01246$ |
| | Known mean reference | $0.33349 \pm 0.03061$ | $0.26183 \pm 0.01159$ |
| Lognormal (clean) | Sample mean | $0.35469 \pm 0.03829$ | $0.28321 \pm 0.01394$ |
| | MOM estimator | $0.35436 \pm 0.03762$ | $0.28302 \pm 0.01370$ |
| | Known mean reference | $0.34924 \pm 0.03677$ | $0.27681 \pm 0.01126$ |
| Gaussian (contaminated) | Sample mean | $0.57954 \pm 0.08882$ | $0.36766 \pm 0.03998$ |
| | MOM estimator | $0.36782 \pm 0.05044$ | $0.27283 \pm 0.01443$ |
| | Known mean reference | $0.33963 \pm 0.03639$ | $0.26294 \pm 0.01076$ |
| Student-$t(5)$ (contaminated) | Sample mean | $0.57001 \pm 0.10104$ | $0.36172 \pm 0.04740$ |
| | MOM estimator | $0.36058 \pm 0.04659$ | $0.27291 \pm 0.01622$ |
| | Known mean reference | $0.34051 \pm 0.03993$ | $0.26246 \pm 0.01278$ |
| Lognormal (contaminated) | Sample mean | $0.59532 \pm 0.12826$ | $0.39235 \pm 0.04752$ |
| | MOM estimator | $0.41233 \pm 0.08114$ | $0.29351 \pm 0.01816$ |
| | Known mean reference | $0.39553 \pm 0.07049$ | $0.28416 \pm 0.01659$ |

stress tests beyond the clean i.i.d. mean-estimation guarantee above, but they support the same operational message: QTES is not intrinsically fragile to an unknown location, provided the foldwise centering step is not itself fragile.

