# OpenReview forum: "Cross-Fitted Clipped Covariance Estimation with a Data-Driven Tail-Energy Criterion"
_TMLR — Under review for TMLR_

### Review · Reviewer_Sx1p · 2026-05-15

**Summary Of Contributions:**

The authors consider how to pick the clipping level for robust covariance estimation. Under the assumptions of known mean and bounded kurtosis, they theoretical characterize a strategy using cross-validation that bounds the bias from clipping using an estimatable "tail energy", making the variance-bias tradeoff computable. They show this strategy is highly competitive on simulated data with different characteristics (different heavy tail distributions, contamination), even relative to an oracle in some settings.

**Audience:**

Yes

**Audience Explanation:**

My main comment is that the manuscript is very dense. Robust covariance estimation is an important task (as the authors point out!) and this work could be made much more accessible to the typical practicioner with a more gentle intro. Set up the problem, explain how heavy-tails/outliers/contamination will break standard covariance estimators, introduce clipping as a solution, with the caveat of "how much do we clip?", then dive into the math. also define every acronym the first time it is used.

**Claims And Evidence:**

Yes

**Claims Explanation:**

I am not a theorist (so I a bit surprised I got this paper to review) but from my non-thorough reading the theoretical results seem rigorous and practical. As a minor comment it would be helpful to understand how much unknown mean messes things up.

**Requested Changes:**

More accessible introduction/exposition.

---

> ### Author Response · Authors · 2026-05-31
>
> We thank the reviewer for the helpful comments. We have substantially revised the introduction to make the exposition more accessible to a broader TMLR audience. The revised introduction now starts from the basic covariance-estimation problem, explains why heavy tails and contamination can make the sample covariance unstable, introduces radial clipping as a simple robustification device, and then frames the main problem as choosing the clipping level, namely the bias--variance tradeoff induced by clipping. We have also checked the manuscript and define each acronym at first use.
>
> We also thank the reviewer for the comment about the known-mean assumption. In response, we added a dedicated appendix on the unknown-mean setting. The core point is algebraic. Let $\\hat\\mu^{(k)}$ be a mean estimator computed only from the training fold, define
>
> $$
> \\Delta_k=\\hat\\mu^{(k)}-\\mu,\\qquad
> V_i^{(k)}=X_i-\\hat\\mu^{(k)}=W_i-\\Delta_k .
> $$
>
> Conditional on the training-fold information $\\mathcal F_k$, both $\\Delta_k$ and the clipping radius $\\tau_k(\\gamma)$ are fixed. Therefore, the validation-fold calibration argument still applies after replacing the centered observations by $V_i^{(k)}$.
>
> The clipping bias decomposes as follows. With $\\alpha_i^{(k)}=\\alpha(V_i^{(k)})$,
>
> $$
> \\begin{aligned}
> &\\Sigma-\\mathrm{E}\\left[(\\alpha_i^{(k)})^2 V_i^{(k)}(V_i^{(k)})^T \\mid \\mathcal F_k\\right] \\\\
> &\\quad =
> \\mathrm{E}\\left[(1-(\\alpha_i^{(k)})^2)V_i^{(k)}(V_i^{(k)})^T \\mid \\mathcal F_k\\right]
> -\\Delta_k\\Delta_k^T .
> \\end{aligned}
> $$
>
> Consequently,
>
> $$
> \\lVert \\mathrm{bias} \\rVert_{\\mathrm{op}}
> \\le
> \\mathrm{E}\\left[
> \\left(\\lVert V_i^{(k)} \\rVert_2^2 - \\tau_k(\\gamma)^2\\right)_+
> \\mid \\mathcal F_k
> \\right]
> +
> \\lVert \\Delta_k \\rVert_2^2 .
> $$
>
> The first term is exactly the QTES tail-energy term, now computed after foldwise centering. The additional term $\\|\\Delta_k\\|_2^2$ is independent of the clipping hyperparameter $\\gamma$. Thus, an unknown mean raises the overall error bound through the usual centering-estimation price, but it does not introduce a new clipping-level calibration difficulty.
>
> We also supplemented the appendix with numerical simulations under a nonzero mean, using two common foldwise centering rules: the sample mean and the geometric median-of-means (MOM) estimator. The results show that estimating the mean changes performance only mildly in clean settings. Under contamination, the ordinary sample mean can be pulled by the inflated observations, while the MOM centering step largely removes this deterioration and stays close to the known-mean reference. Some representative results are shown below; the full table is included in the revised appendix.
>
> | Scenario | Mean estimator | RelSpecErr | SubspaceErr |
> |---|---|---:|---:|
> | Gaussian clean | Sample mean | $0.34991\\pm0.03777$ | $0.26875\\pm0.01378$ |
> | Gaussian clean | MOM estimator | $0.35049\\pm0.03770$ | $0.26888\\pm0.01378$ |
> | Gaussian clean | Known mean reference | $0.33300\\pm0.03177$ | $0.26251\\pm0.01073$ |
> | Lognormal clean | Sample mean | $0.35469\\pm0.03829$ | $0.28321\\pm0.01394$ |
> | Lognormal clean | MOM estimator | $0.35436\\pm0.03762$ | $0.28302\\pm0.01370$ |
> | Lognormal clean | Known mean reference | $0.34924\\pm0.03677$ | $0.27681\\pm0.01126$ |
> | Gaussian contaminated | Sample mean | $0.57954\\pm0.08882$ | $0.36766\\pm0.03998$ |
> | Gaussian contaminated | MOM estimator | $0.36782\\pm0.05044$ | $0.27283\\pm0.01443$ |
> | Gaussian contaminated | Known mean reference | $0.33963\\pm0.03639$ | $0.26294\\pm0.01076$ |
> | Student-$t(5)$ contaminated | Sample mean | $0.57001\\pm0.10104$ | $0.36172\\pm0.04740$ |
> | Student-$t(5)$ contaminated | MOM estimator | $0.36058\\pm0.04659$ | $0.27291\\pm0.01622$ |
> | Student-$t(5)$ contaminated | Known mean reference | $0.34051\\pm0.03993$ | $0.26246\\pm0.01278$ |

---

### Review · Reviewer_17qY · 2026-06-01

**Summary Of Contributions:**

The manuscript presents a method to robustly estimate covariance in heavy-tailed distributions, for which this estimation is unstable. The proposed solution is to radially clip the observations to make the estimation robust and a data driven way to tune radial clip level tau. The data driven method is: scan several tau values and estimate variance and clipping bias, return the best.

Strength
* linking tail energy to covariance clipping bias

Weaknesses:
* Paper clarity. The whole manuscript is unnecessarily convoluted.
* The guarantees (theorem 3 and corollary 4) seem about choosing a good clipping level within the algorithm’s clipping family (proposed bias variance criterion). In this sense the proposed method is not a SOTA provenly optimal covariance estimator, as stated by the authors themselves, and the contribution seems limited.
* The results are encouraging but not decisive, they show that QETS is a plausible heuristic/tuning rule. A comparison with more recent estimators or online estimators, like an online quantile based clipping, is appropriate.


Minor:
* Sentences like “unusually transparent” “hidden scale knowledge” “largely settle what is statistically possible” “On the sharp
statistical front” don’t help understanding for the average reader.
* “Because these estimator families share the same structural foundation, evaluating the Wei–Minsker estimator at the analytically optimal scale derived from the unobservable true distribution provides a formidable, theory-guided oracle baseline.” ?
* “Structural foundation” what do you mean? “At the analyitically optimal scale” what’s the scale you are referring to? “Derived from the unobservable true distribution” how do you derive from the unobservable distribution, if it is, indeed, unobservable?
* “By collapsing its candidate grid to this single exact point, we completely disable its Lepski adaptation procedure, yielding a privileged non-data-driven reference.“
“To account for the split randomness, we define the filtration…independent”. Not clear what was the point of this passage.
“The positive grid is...” The grid is defined by rho^j, which are not uniquely defined. Please explain clearly this point.
If you used just 2 sets, why nret := 4⌊n/4⌋, can’t you nret := 2⌊n/2⌋? The overall description of a 2-folds cross validation seems a bit convoluted.

**Audience:**

No

**Audience Explanation:**

The core of the manuscript is an heuristic to robustly estimate the covariance, it remains of little interest without a sound comparison with SOTA algorithms

**Claims And Evidence:**

Yes

**Claims Explanation:**

Derivations seem correct, as well as the numerical comparison.

**Requested Changes:**

* Rewrite the manuscript in a more accessible, direct language. At each point explain in plain English what are you going to do.
* Compare with SOTA algorithms for covariance estimation and online estimators.

---

> ### Author Response · Authors · 2026-06-03
> **Part 1**
>
> We thank the reviewer for the careful reading and for pointing out that the previous manuscript was harder to read than necessary. We have substantially rewritten the paper in a more direct style. The introduction now states the scope of the contribution before technical details. Section 2 has been reorganized around the bias--variance tradeoff induced by clipping: Section 2.1 explains the clipped covariance family and the variance certificate, Section 2.2 explains how tail energy controls clipping bias, and Section 2.3 combines these two terms into the QTES selection rule. Before the algorithm, we now give a plain-English summary of the complete procedure. We also removed or rewrote the phrases mentioned by the reviewer.
>
> We also clarified the bookkeeping in the algorithm. The filtration is introduced for only one reason: after conditioning on the training fold, the clipping radius is fixed, while the held-out observations remain independent; this is what allows the matrix concentration bound to be applied. We now state this explicitly. The grid is now defined by
>
> $$
> \\begin{aligned}
> J_{\\max}
> &=
> \\left\\lfloor
> \\frac{\\log(\\gamma_{\\max}/\\gamma_{\\min})}
> {\\log\\rho}
> \\right\\rfloor, \\\\
> \\mathcal G
> &=
> \\{\\gamma_j=\\gamma_{\\max}\\rho^{-j}:j=0,\\ldots,J_{\\max}\\}.
> \\end{aligned}
> $$
>
> Thus the grid is uniquely determined once $\\rho$, $\\gamma_{\\min}$, and $\\gamma_{\\max}$ are fixed. In the experiments we use the dyadic choice $\\rho=2$, matching the multiplicative grid spacing used by the Wei--Minsker Lepski selector. We also explain why $n_{\\mathrm{ret}}=4\\lfloor n/4\\rfloor$: although the procedure uses two folds, the variance proxy pairs observations inside each held-out fold, so each fold must have even size. This is why retaining a multiple of four is used rather than $2\\lfloor n/2\\rfloor$.
>
> We strengthened the comparison with theoretically state-of-the-art truncation methods. Wei--Minsker (2017) [1] is the most relevant optimal-rate reference for our setting because its single-scale estimator is exactly the Euclidean radial clipping family studied here. In the known-mean case, the Wei--Minsker estimator can be written as
>
> $$
> \\hat\\Sigma_{\\theta}=\\frac{1}{m}\\sum_{i=1}^m
> Z_iZ_i^T
> \\min\\left\\{1,
> \\frac{1}{\\theta\\lVert Z_i\\rVert_2^2}
> \\right\\}.
> $$
>
> Setting $\\tau^2=1/\\theta$ and
>
> $$
> \\widetilde Z_i(\\tau)=Z_i\\min\\left\\{1,
> \\frac{\\tau}{\\lVert Z_i\\rVert_2}
> \\right\\},
> $$
>
> we have
>
> $$
> \begin{aligned}
> \\widetilde Z_i(\\tau)\\widetilde Z_i(\\tau)^T=Z_iZ_i^T
> \\min\\left\\{1,
> \\frac{\\tau^2}{\\lVert Z_i\\rVert_2^2}
> \\right\\} \\\\=
> Z_iZ_i^T
> \\min\\left\\{1,
> \\frac{1}{\\theta\\lVert Z_i\\rVert_2^2}
> \\right\\}.
> \end{aligned}
> $$
>
> Under the required preliminary scale interval
>
> $$
> \\sigma_{\\min}
> \\le
> \\sigma_0
> \\le
> \\sigma_{\\max},
> \\sigma_0=\\left\\lVert
> \\mathrm{E}\\left[
> \\lVert Z\\rVert_2^2ZZ^T
> \\right]
> \\right\\rVert_{\\mathrm{op}}^{1/2},
> $$
>
> Corollary 2.2 of Wei--Minsker (2017) gives, up to constants and the $L_4$--$L_2$ moment factor, the effective-rank operator-norm rate
>
> $$
> \\begin{aligned}
> \\left\\lVert \\hat\\Sigma_{\\mathrm{WM}}-\\Sigma\\right\\rVert_{\\mathrm{op}}
> &\\le
> C\\lVert\\Sigma\\rVert_{\\mathrm{op}}
> \\sqrt{\\frac{r(\\Sigma)\\beta}{m}}, \\\\
> r(\\Sigma)
> &=
> \\frac{\\mathrm{tr}(\\Sigma)}
> {\\lVert\\Sigma\\rVert_{\\mathrm{op}}}.
> \\end{aligned}
> $$
>
> We therefore added two Wei--Minsker references.
> WM-adaptive is the full Lepski algorithm and should be viewed as the theoretically SOTA-level Euclidean truncation comparator, modulo the required prior scale interval.
> WM-oracle is deliberately even more favorable than a standard data-driven SOTA comparison: because the simulation distribution is known, we compute the unknown scale $\\sigma_0$ from the data-generating model and set $\\sigma_{\\min}=\\sigma_{\\max}=\\sigma_0$. This is explicit oracle data leakage. It collapses the Lepski grid to a single scale and disables the adaptive step, giving Wei--Minsker information that would not be available in real data. Thus WM-oracle is a stronger-than-SOTA oracle reference for the same Euclidean clipping family.
>
> As a complementary practical data-driven comparison, we also added the Python package `tfHuber` by Dai and Sun (2021) [2], whose covariance routines are motivated by the user-friendly covariance estimation framework of Ke et al. (2019) [3]. Unlike Wei--Minsker, `tfHuber` does not require a user-supplied preliminary interval $[\\sigma_{\\min},\\sigma_{\\max}]$ for the unknown matrix-variance scale. Although this line of Huber-type covariance estimation is not the effective-rank optimal benchmark represented by Wei--Minsker, it is a recent, computable, tuning-light robust covariance method and is therefore an appropriate practical baseline.

---

> ### Author Response · Authors · 2026-06-03
> **Part 2**
>
> We also considered the suggested online quantile-clipping direction. We did not find a canonical, widely used Python implementation or standard benchmark for this specific online covariance estimator; implementing one would require additional choices of streaming quantile sketch, update schedule, and clipping rule. We therefore avoided introducing a hand-tuned ad hoc baseline and instead added reproducible computable baselines with clear references.
>
> The condensed tables below focus on QTES and the newly added Wei--Minsker and tfHuber comparisons.
>
> The revised experiments include the following additional comparisons. For the clean-data benchmark at $(n,d)=(400,200)$, entries report relative spectral error and runtime in seconds. `tfHuber` rows use three trials due to runtime; all other rows use 100 trials.
>
> | Method | Gaussian | Student-$t(5)$ | Lognormal |
> |---|---:|---:|---:|
> | QTES | 0.3330, 0.030s | 0.3335, 0.030s | 0.3492, 0.027s |
> | WM-oracle | 0.3487, 0.001s | 0.3451, 0.001s | 0.3417, 0.001s |
> | WM-adaptive | 0.5029, 0.005s | 0.4952, 0.011s | 0.4462, 0.012s |
> | `tfHuber`-element | 0.3133, 319s | 0.3593, 346s | 0.3248, 418s |
> | `tfHuber`-spectrum | 0.3239, 528s | 0.3803, 504s | 0.3852, 515s |
>
> For the gross-contamination benchmark, entries report relative spectral errors under standardized lognormal inliers with 5% inflated Gaussian outliers.
>
> | Method | $c=1$ | $c=25$ | $c=100$ | $c=400$ |
> |---|---:|---:|---:|---:|
> | QTES | 0.3484 | 0.3833 | 0.3911 | 0.3873 |
> | WM-oracle | 0.3406 | 0.3563 | 0.3549 | 0.3547 |
> | WM-adaptive | 0.4462 | 0.4424 | 0.4455 | 0.4460 |
> | `tfHuber`-element | 0.3124 | 2.1182 | 7.6333 | 29.6632 |
> | `tfHuber`-spectrum | 0.4047 | 1.0931 | 0.9105 | 0.8989 |
>
> These results show that QTES is competitive with the scale-leaked WM-oracle on clean data, consistently improves over the full adaptive Wei--Minsker implementation, and remains more stable than `tfHuber` under large contamination. Taken together, they suggest that QTES is a strong practical robust covariance estimator when the data distribution is unknown and the presence or strength of contamination is unclear. The comparison also illustrates an important practical point: an optimal effective-rank convergence rate does not by itself guarantee the best finite-sample performance. WM-adaptive has a theoretically optimal-rate guarantee under its assumptions, but its empirical performance is poor in our finite-sample experiments. WM-oracle performs better only after receiving explicit oracle data leakage. QTES represents a tradeoff: it gives up a direct optimal-rate guarantee, but it is fully data-driven, does not require a prior scale interval, and shows stable empirical performance across clean heavy-tailed and contaminated settings.
>
>
> ## References
>
> [1] Xiaohan Wei and Stanislav Minsker. 2017. *Estimation of the Covariance Structure of Heavy-Tailed Distributions*. Advances in Neural Information Processing Systems 30, 2859--2868.
>
> [2] Yifan Dai and Qiang Sun. 2021. *tfHuber: Python package for tuning-free Huber estimation and regression*, Version 0.1.1. https://pypi.org/project/tfHuber/
>
> [3] Yuan Ke, Stanislav Minsker, Zhao Ren, Qiang Sun, and Wen-Xin Zhou. 2019. *User-Friendly Covariance Estimation for Heavy-Tailed Distributions*. Statistical Science 34(3), 454--471.

---

### Review · Reviewer_MnLk · 2026-06-08

**Summary Of Contributions:**

The paper studies data-driven calibration of a centered, clipped covariance estimator for heavy-tailed data in the known-mean setting. The main idea of the authors is that, for Euclidean radial clipping, the operator-norm bias is bounded by a scalar tail-energy quantity. The proposed Quantile Tail-Energy Surrogate (QTES) method selects the clipping level over a geometric grid. Under a finite fourth moment assumption and a block feasibility condition, the authors prove a uniform finite-sample calibration bound and a near-oracle guarantee relative to the best clipping level on the grid for the same bias/variance criterion. Experiments on synthetic heavy-tailed and contaminated spiked covariance models show that QTES improves over uninformed clipping in several settings. Overall, the contribution is a transparent and implementable tuning rule for a simple clipped covariance family.

**Audience:**

Yes

**Audience Explanation:**

I think the paper is a bit borderline in this regard as it does not provide significant new theory, and the empirical evidence is somewhat weak. See the Requested CHanges for ways I think this could be strengthened.

**Broader Impact Concerns:**

This paper is primarily theoretical in nature, with the main contribution being a general algorithm for robust covariance estimation under heavy-tailed conditions. As such, the paper does not require a Broader Impact Statement.

**Claims And Evidence:**

Yes

**Claims Explanation:**

The paper is very clear about the claims it makes and the assumptions upon which they are based. The author provide theoretical and empirical evidence for all claims.

**Requested Changes:**

The authors clearly position their contribution in the *calibration* regime (i.e. a practical and robust method for covariance estimation) and not as a new rate-optimal theory. Thus, I think the empirical validation could be strengthened by adding a broader range of data driven methods. For example:
- The [Wei & Minsker, NeurIPS 2017] equipped with Lepski's method as proposed in the original paper, instead of just with an oracle upper bound.
- The User-friendly method of [Ke et al, Statistical Science 2019]. The manuscript specifically positions itself similarly to the "automatic calibration" viewpoint of Ke et al., this seems like a natural comparison.
- The Robust U-Statistic method of [Minsker and Wei, Bernoulli 2020].

Expanding the empirical validation with these (and possibly others I am unaware of) would strenghthen the author claims of practicality of the proposed approach.

---

> ### Author Response · Authors · 2026-06-11
>
> We thank the reviewer for the careful and constructive suggestions. We agree that, since our contribution is positioned as a practical calibration method rather than a new rate-optimal covariance estimator, broader data-driven comparisons are important.
>
> In response, we expanded the empirical section to include all three suggested lines of work. First, we added the original Lepski-type Wei--Minsker procedure as WM-adaptive. This implementation follows the full Lepski scale-selection step, while giving it the favorable scale bracket $[\\sigma_0/2, 2 \\sigma_0]$, which contains the true $\sigma_0$.
>
> Second, we added `tfHuber`, whose covariance estimator implements the tuning-free/user-friendly Huber covariance approach of Ke et al. (2019). We report both elementwise and spectrumwise covariance modes.
>
> Third, we added Minsker--Wei (2020) MW-Ustat benchmarks: an oracle-scale version using the true $\sigma_* $ and an adaptive version with $\sigma_{min} = \sigma_*/2$. For computational feasibility, the adaptive grid is restricted to the three scales ${\sigma_*/2, \sigma_*, 2 \sigma_*}$. This grid contains the oracle scale and its immediate lower and upper neighbors, which is sufficient for testing Lepski selection near the theoretically relevant scale in our diagnostic comparison, while avoiding the substantial extra cost of running the iterative MW-Ustat estimator on a wider grid.
>
>
> ## references
>
> Xiaohan Wei and Stanislav Minsker. 2017. Estimation of the Covariance Structure of Heavy-Tailed Distributions. Advances in Neural Information Processing Systems 30, 2859--2868.
>
> Yifan Dai and Qiang Sun. 2021. tfHuber: Python package for tuning-free Huber estimation and regression, Version 0.1.1. https://pypi.org/project/tfHuber/
>
> Yuan Ke, Stanislav Minsker, Zhao Ren, Qiang Sun, and Wen-Xin Zhou. 2019. User-Friendly Covariance Estimation for Heavy-Tailed Distributions. Statistical Science 34(3), 454--471.
>
> Stanislav Minsker and Xiaohan Wei. 2020. Robust Modifications of U-Statistics and Applications to Covariance Estimation Problems. Bernoulli 26(1), 694--727.

---

### Author Response · Authors · 2026-06-11
**Overall Response to the Reviewers**

We thank all three reviewers for their careful and constructive comments. We have adjusted it accordingly and believe it is much better with your feedback. In the revised manuscript, we made the following main changes：

We substantially rewrote the narrative and exposition, following the comments of Reviewers Sx1p and 17qY.

We expanded the empirical comparisons, as requested by Reviewers MnLk and 17qY.

We added an unknown-mean extension, addressing Reviewer Sx1p’s comment.